# ONE-TOKEN VERIFICATION FOR REASONING LLMS, ANYTIME, ANYWHERE

## ABSTRACT

Reasoning large language models (LLMs) have recently achieved breakthrough performance on complex tasks such as mathematical problem solving. A widely used strategy to further improve performance is parallel thinking, wherein multiple reasoning traces are generated, and the final prediction is chosen using methods such as Best-of-$N$ or majority voting. However, two limitations remain: most existing methods lack effective mechanisms for assessing the quality of reasoning traces, typically relying only on final logits or answers, and multi-sample decoding incurs substantial inference latency for long outputs. To address these challenges, we propose *One-Token Verification* (OTV), a lightweight framework for assessing reasoning quality via a single-token forward during generation. The proposed OTV method introduces a learnable LoRA-based role vector that, without interfering with the primary reasoning process, enables the LLM to assume a verification role and probe the past KV cache for correctness confidence estimation. Unlike generic verifiers or external reward models, OTV is trained natively for each LLM, directly leveraging internal computations for token-level scoring. As a result, OTV can provide confidence signals at any point in generation and for any token position, realizing "anytime, anywhere" verification. Experiments on math benchmarks show that OTV consistently outperforms the state-of-the-art baselines in parallel thinking. Moreover, based on OTV, we introduce efficient variants that terminate most traces early and retain only a single complete reasoning path, reducing token usage by up to 90%. In this setting, OTV maintains superior performance while favoring shorter and more reliable solutions.

## 1 INTRODUCTION

Large language models (LLMs) such as OpenAI o1 (Jaech et al., 2024), DeepSeek-R1 (Guo et al., 2025), and the Qwen3 series (Yang et al., 2025a) have recently demonstrated strong multi-step reasoning capabilities in mathematical problem solving, supported by increasingly mature training pipelines that incorporate long chain-of-thought (CoT) fine-tuning (Wei et al., 2022; Suzgun et al., 2023) and reinforcement learning (Ouyang et al., 2022; Shao et al., 2024; Team et al., 2025).

To further improve reasoning performance, a complementary line of work scales computation at inference time, also referred to as test-time scaling (Brown et al., 2024; Zhang et al., 2025b; Venktesh et al., 2025). A common strategy is to generate multiple solution attempts in parallel and then select the final response based on the quality of the reasoning trace. Existing approaches fall into two broad categories: methods that rely on the model's own token distribution or calibration (Wang et al., 2022; Kang et al., 2025; Fu et al., 2025; Zhang et al., 2025a; Huang et al., 2025), and methods that depend on external verifiers (Cobbe et al., 2021; Hosseini et al., 2024; Zhang et al., 2024c), reward models (Lightman et al., 2023; Wang et al., 2024; Yang et al., 2024), or aggregators (Zhao et al., 2025). The key distinction between the two categories lies in whether a general external model is used to assess answers. Self-reflection or self-verification is more lightweight but suffers from poor calibration (Huang et al., 2023; Xiong et al., 2024), whereas external verifiers provide stronger supervision with limited generalization. Although these strategies narrow the gap between average accuracy and the upper bound given by Pass@$k$ (Chen et al., 2021), substantial discrepancies remain. Moreover, the cost of full-sequence generation, together with "System-2" style overthinking (Chen et al., 2024a), raises practical concerns. These limitations highlight the need for an efficient and reliable verification mechanism that leverages internal signals without incurring heavy computation.

Figure 1: Motivation of One-Token Verification. OTV verifier probes internal dynamics (KV cache) to estimate correctness during inference, analogous to how inspector judges reasoning step by step.

To this end, we propose *One-Token Verification* (OTV), illustrated in Figure 1. To analyze a model's evolving confidence in the correctness of its answers during the generation, we augment a reasoning model (e.g., Qwen3-8B) with a lightweight LoRA module, yielding a variant such as Qwen3-8B-OTV as a native verifier that shares the same architecture as the base model. A key insight is that the key–value (KV) cache preserves all intermediate states and memory accumulated during reasoning. OTV introduces a cross-attention mechanism where a special token, equipped with the LoRA module, attends to the reused KV cache from the base model (without LoRA). This design allows the verifier to extract signals from the reasoning process and estimate token-level correctness confidence[1] at any step, without disrupting the primary reasoning flow. Recent studies suggest that internal representations of LLMs already encode correctness-related signals, sometimes even before decoding completes (Burns et al., 2022; Azaria & Mitchell, 2023; Zhang et al., 2025a; Li et al., 2025), which motivates our use of the KV cache as a richer source for verification. In doing so, OTV equips LLM with a native verifier that explicitly assesses the quality of intermediate reasoning at the token level during generation, in contrast to external verifiers that operate only on outputs.

Our contributions are as follows: (i) We propose a lightweight LoRA-based verifier that reuses the base model's KV cache to deliver anytime, token-level confidence estimation with only a single one-token forward; (ii) We design a heuristic supervision and parallel training scheme that provides dense step-level signals, making the proposed OTV method efficient, scalable, and adaptable to metrics beyond answer correctness; (iii) We demonstrate that OTV consistently outperforms prior baselines across multiple reasoning LLMs and math benchmarks.

## 2 RELATED WORK

**Parallel thinking** Parallel thinking has emerged as a prominent test-time scaling strategy (Comanici et al., 2025; Wen et al., 2025; Yang et al., 2025b; Hsu et al., 2025; Zheng et al., 2025). Unlike extended thinking (Jaech et al., 2024; Muennighoff et al., 2025), which increases depth by producing longer chains, it samples multiple reasoning traces and selects among them. Typical methods include Best-of-$N$ (Stiennon et al., 2020; Gao et al., 2023), majority voting (Wang et al., 2022), and tree-based methods (Yao et al., 2023; Zhang et al., 2024b), where traces are scored by internal or external evaluators. Parallel thinking generally achieves a higher performance ceiling (Ghosal et al., 2025), but incurs substantial token overhead. For budget controlling, recent work explores token-supervised value models (Lee et al., 2025) and trajectory pruning (Wang et al., 2025; Fu et al., 2025; Huang et al., 2025) to assess step-level quality and stop low-quality traces early.

**Reasoning trace quality assessment** Assessing reasoning trace quality (Lee & Hockenmaier, 2025; Venktesh et al., 2025) is critical for enhancing the reliability of LLMs in parallel thinking. Existing approaches fall into external and internal verification. External methods train auxiliary verifiers to evaluate either final outcomes or intermediate steps: outcome reward models (ORMs) judge only the correctness of the final answer (Cobbe et al., 2021; Yu et al., 2023a; Chen et al., 2024b;

---

[1]Correctness confidence in this paper denotes the estimated probability that the reasoning trace is correct.

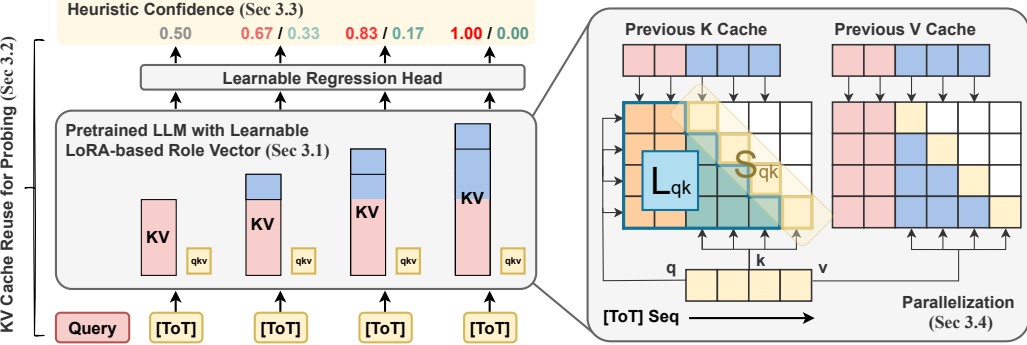

Figure 2: Overview of the One-Token Verification (OTV) framework and its four components.

Liu et al., 2024; Lu et al., 2024; Zhang et al., 2025c), whereas process reward models (PRMs) assign scores to reasoning steps and aggregate them into answer-level estimates (Uesato et al., 2022; Lightman et al., 2023; Wang et al., 2023; Zhang et al., 2025d). Extensions broaden verification across domains (e.g., VersaPRM (Zeng et al., 2025)) and criteria (Golovneva et al., 2022; Wang et al., 2024), while others recast verification as generative prediction (e.g., CLoud (Ankner et al., 2024), GenRM (Zhang et al., 2024c)), introduce critic-style feedback models (Zheng et al., 2023; 2024a; Ye et al., 2025), or aggregate from multiple verifiers and solutions to improve the evaluation accuracy (Lifshitz et al., 2025; Zhong et al., 2025; Zhao et al., 2025).

Internal verification methods exploit the uncertainty (Lin et al., 2022; Fadeeva et al., 2024) and information encoded within the model itself to assess the reasoning reliability. Self-consistency (Wang et al., 2022) aggregates diverse traces via majority voting, while self-certainty (Kang et al., 2025) quantifies quality from output distributions. Confidence-aware approaches such as DeepConf (Fu et al., 2025) prune low-quality traces using token-level signals, and self-calibration (Huang et al., 2025) distills confidence back for one-pass estimation. Probing hidden states (Zhang et al., 2025a) further shows that models internally encode correctness signals that enable calibrated early exits.

Overall, external verifiers provide robust supervision but ignore internal reasoning dynamics and add extra costs, while internal methods are lightweight but often poorly calibrated. Our approach lies between them: a LoRA module enables the model to switch into a verifier role, with OTV operating directly on the internal KV cache for reliable confidence estimation with negligible overhead.

## 3 ONE-TOKEN VERIFICATION FOR REASONING LLMS

We now introduce the *One-Token Verification* (OTV) framework. As shown in Figure 2, OTV integrates a reasoning LLM with a LoRA-based role vector and a special verification token that probes the KV cache, whose hidden states are mapped through a regression head into confidence scores supervised by heuristic labels. We next describe its four components: role vector design (Section 3.1), KV cache probing (Section 3.2), heuristic confidence (Section 3.3), and parallelization (Section 3.4).

### 3.1 LORA AS ROLE VECTOR FOR SELF-VERIFICATION

Low-rank adaptation (LoRA) (Hu et al., 2022) is a widely used parameter-efficient fine-tuning technique that adapts a pretrained model by injecting trainable low-rank matrices into the weight updates of linear layers. Given a weight matrix $W_0 \in \mathbb{R}^{d_2 \times d_1}$, LoRA uses matrices $A \in \mathbb{R}^{r \times d_1}$ and $B \in \mathbb{R}^{d_2 \times r}$ with rank $r \ll \min\{d_1, d_2\}$ such that the updated weight is $W = W_0 + \Delta W = W_0 + BA$. During training, $W_0$ remains frozen, leading to significant savings in both memory and computation.

In our framework, LoRA is used not only for efficient adaptation but also as a *role vector* for self-verification. To preserve the foundation model's reasoning ability, we adopt a gating mechanism inspired by Samragh et al. (2025). The LoRA path is added in parallel to the original linear layers and activated only in the verification mode. For an input $x_t \in \mathbb{R}^{d_1}$ at position $t$, the output is

$$y_t = W_0 x_t + I(t) BA x_t \in \mathbb{R}^{d_2}, \tag{1}$$

where $I(t) \in \{0, 1\}$ is a gate variable. When $I(t) = 0$, the model functions as the original reasoner; when $I(t) = 1$, the verifier role is engaged. This gating enables seamless switching between reasoning and verification, supporting self-assessment with minimal risk of degrading the base model.

## 3.2 KV Cache Reuse for Probing Reasoning Dynamics

During decoding, transformer-based LLMs maintain a *key–value (KV) cache*, which stores the intermediate representations from all steps. Let the cache at the $(t-1)$-th token and layer $\ell$ be

$$C_{t-1}^{(\ell)} = (K_{t-1}^{(\ell)}, V_{t-1}^{(\ell)}), \quad K_{t-1}^{(\ell)} = [k_1^{(\ell)}, \ldots, k_{t-1}^{(\ell)}], \ V_{t-1}^{(\ell)} = [v_1^{(\ell)}, \ldots, v_{t-1}^{(\ell)}], \tag{2}$$

where $k_i^{(\ell)}, v_i^{(\ell)} \in \mathbb{R}^d$ denote the key and value vectors of the $i$-th token at layer $\ell$. The next-token prediction at step $t$ for an $L$-layer LLM can thus be expressed as a function of the current input $x_t$, the parameters, and the KV cache $C_{t-1}$. Compared with hidden states, which capture local information at individual layers and steps, the KV cache provides a complete record of the attention context across all layers and the entire reasoning trajectory. Thus, it captures richer signals of the model's internal computation, including implicit confidence in the correctness of its ongoing solution.

To exploit this property, we design a verification mechanism that reuses the KV cache together with the LoRA-based role vector. A special verification token, the *token of truth* ([ToT]), is inserted in the verification mode (i.e., $I(t) = 1$) at any position $t$ to estimate the correctness confidence up to $(t-1)$-th token. Rather than recomputing the prefix, the model reuses the KV cache $C_{t-1}^{(\ell)}$ from each layer $\ell$ and performs a forward pass with LoRA modules. For layer $\ell$, let $x_t^{(\ell)}$ denote the input ($x_t^{(0)}$ being the embedding). The LoRA-updated query, key, and value vectors are computed as

$$\tilde{q}_t^{(\ell)} = \big(W_{0,q}^{(\ell)} + B_q^{(\ell)} A_q^{(\ell)}\big)x_t^{(\ell)}, \ \tilde{k}_t^{(\ell)} = \big(W_{0,k}^{(\ell)} + B_k^{(\ell)} A_k^{(\ell)}\big)x_t^{(\ell)}, \ \tilde{v}_t^{(\ell)} = \big(W_{0,v}^{(\ell)} + B_v^{(\ell)} A_v^{(\ell)}\big)x_t^{(\ell)}, \tag{3}$$

where $W_{0,*}^{(\ell)}$ denotes pretrained weights for the corresponding component at layer $\ell$ and $A_*^{(\ell)}, B_*^{(\ell)}$ are trainable low-rank matrices in LoRA for the corresponding component at layer $\ell$. The attention for [ToT] at layer $\ell$ is computed as

$$\text{Attn}(x_t^{(\ell)}) = \text{softmax}\left(\frac{\tilde{q}_t^{(\ell)}\big[K_{t-1}^{(\ell)}, \tilde{k}_t^{(\ell)}\big]^\top}{\sqrt{d}}\right)\big[V_{t-1}^{(\ell)}, \tilde{v}_t^{(\ell)}\big]. \tag{4}$$

After [ToT] is propagated through all layers, its final hidden state $h_{\text{ToT}} \in \mathbb{R}^d$ is obtained. Rather than decoding into the vocabulary space, a lightweight regression head $g$ (with a three-layer neural network ) maps $h_{\text{ToT}}$ to a scalar confidence score as

$$c_t = g(h_{\text{ToT}}) \in [0, 1],$$

which serves as an estimate of the correctness of the reasoning up to the $t$-th token.

## 3.3 Heuristic Confidence for Supervised Alignment

Unlike computationally expensive approaches based on PRM datasets (Lightman et al., 2023) or MCTS-based sampling (Wang et al., 2023; Luo et al., 2024; Zhang et al., 2024a; Feng et al., 2024; Setlur et al., 2024; Guan et al., 2025), we employ a heuristic strategy that assigns pseudo-confidence values to all tokens. Given a calibration dataset, we sample model outputs and denote each reasoning trace as $(x_1, \ldots, x_T)$. For the $t$-th token, we assign a correctness confidence $c_t \in [0, 1]$, where 0 represents incorrect reasoning, 0.5 denotes maximal uncertainty, and 1 indicates correct reasoning. The ground-truth trajectory $(c_1, \ldots, c_T)$ is determined by the final correctness of the trace and instantiated heuristically. As illustrated in Figure 3, four heuristic schemes capture different intuitions about how confidence should evolve during reasoning.

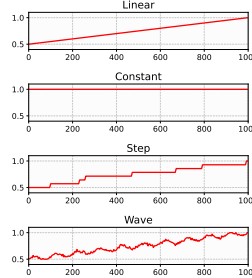

Figure 3: Four heuristic confidence schemes over a 1000-token correct reasoning trace. For incorrect one, the $y$-axis is flipped so that the confidence eventually decays to 0.

- *Linear*: changes linearly from 0.5 toward 1.0 (or 0).

- *Constant*: remains fixed at 1 (or 0).

- *Step*: divides the reasoning trace into segments (using "\n\n" as separators) and changes uniformly.

- *Wave*: follows a linear trend with small sinusoidal fluctuations and Gaussian noise.

After assigning the pseudo-confidence as label, we can train both the LoRA modules and the regression head using the mean squared error (MSE) loss between the predicted confidence and the pseudo-confidence. This procedure grounds the verifier in well-defined confidence patterns without requiring explicit annotations. In our experiments, we use the *Linear* as the default setting, while the other three variants are evaluated through ablations in Section 5.1.

### 3.4 PARALLELIZATION

A key advantage of Transformers (Vaswani et al., 2017) is token-level parallelism during training. OTV preserves this property: although confidence estimation is triggered by inserting the special [ToT] token (Eq. (4)), all [ToT] tokens can be computed in parallel within a single forward pass, which will be introduced in the following.

**Definition 1.** (Lower-Left Augmentation) *We define an operator* LLAug *for lower-left augmentation, which embeds an $n \times n$ lower-triangular matrix L into the lower-left block of an $(n+1) \times (n+1)$ matrix as*

$$\text{LLAug}(L) \ = \ \begin{bmatrix} \mathbf{0}_{1 \times n} & \mathbf{0}_{1 \times 1} \\ L & \mathbf{0}_{n \times 1} \end{bmatrix} \ \in \ \mathbb{R}^{(n+1) \times (n+1)}. \tag{5}$$

Consider a prefix of $t$ tokens with cached keys and values $K_t^{(\ell)}, V_t^{(\ell)} \in \mathbb{R}^{t \times d}$ at layer $\ell$. When verification tokens are inserted, the sequence length becomes $t + 1$. We denote the queries, keys, and values of all [ToT] tokens by $\tilde{Q}_{\text{ToT}}^{(\ell)}, \tilde{K}_{\text{ToT}}^{(\ell)}, \tilde{V}_{\text{ToT}}^{(\ell)} \in \mathbb{R}^{(t+1) \times d}$. Their attention weights consist of two components: $L_{qk}^{(\ell)}$, capturing interactions with the cached prefix states via LLAug, and $S_{qk}^{(\ell)}$, a diagonal term enforcing self-interaction among [ToT] positions for consistency. Formally,

$$S^{(\ell)} = \text{softmax}\left( (L_{qk}^{(\ell)} + S_{qk}^{(\ell)})/\sqrt{d} \right) \in \mathbb{R}^{(t+1) \times (t+1)}, \tag{6}$$

where

$$L_{qk}^{(\ell)} = \text{LLAug}\big(\tilde{Q}_{\text{ToT}, \leq t}^{(\ell)} (K_t^{(\ell)})^\top + M\big), \qquad S_{qk}^{(\ell)} = \text{diag}\big(\tilde{Q}_{\text{ToT}}^{(\ell)} (\tilde{K}_{\text{ToT}}^{(\ell)})^\top\big).$$

Here $\text{diag}(\cdot)$ extracts the diagonal elements to form a diagonal matrix, $\tilde{Q}_{\text{ToT}, \leq t}^{(\ell)} \in \mathbb{R}^{t \times d}$ denotes the slice of the first t rows of $\tilde{Q}_{\text{ToT}}^{(\ell)}$, and $M \in \mathbb{R}^{t \times t}$ is the causal mask, with a triangle of $-\infty$ in the upper right and 0's elsewhere. The final attention outputs are computed as:

$$\text{Attn}^{(\ell)}(X_{\text{ToT}}^{(\ell)}) = \big(S^{(\ell)} - \text{diag}(S^{(\ell)})\big) \big[ V_t^{(\ell)}, \mathbf{0}_{d \times 1} \big] + \text{diag}(S^{(\ell)}) \tilde{V}_{\text{ToT}}^{(\ell)}. \tag{7}$$

It is easy to see that the formulation in Eq. (7) is mathematically equivalent to the single-token computation in Eq. (4), and it enables simultaneous confidence estimation for all positions in one forward pass. Since the query part of the prompt (red in Figures 1 and 2) does not belong to the reasoning process, we apply the MSE loss only to [ToT] tokens within the response part (blue).

Finally, we include algorithmic descriptions of the inference and training procedure in Appendix E.

## 4 EXPERIMENTS

### 4.1 EXPERIMENTAL SETUP

We evaluate our framework on three representative reasoning LLMs: *Qwen3-4B-Instruct*, *Qwen3-8B* (in non-thinking modes) (Yang et al., 2025a), and *Qwen-32B-DAPO* (Yu et al., 2025). These models span diverse parameter scales and training paradigms. For supervision, we use the publicly available DAPO17K dataset (Yu et al., 2025) to calibrate and train the LoRA modules together with the regression head. Each training example is sampled 8 times to provide diverse reasoning traces. Evaluation is performed on the AIME benchmark (MAA), and we prepare a pool of 256 complete traces per test problem. All generations across datasets and models use a sampling temperature of 1.0. Fine-tuning is performed using `LlamaFactory` (Zheng et al., 2024b) for 3 epochs with a learning rate of $1e-4$ and a batch size of 128, distributed across 128 GPUs (96 GB each).

**Baselines** We compare OTV against representative methods for assigning confidence or verification scores to reasoning traces. These include **DeepConf** (Fu et al., 2025), a training-free estimator based on token-level log-likelihood; the **generative verifier (GenRM)** (Zhang et al., 2024c), which estimates correctness by predicting the probability of "Yes" versus "No" given an prompt appended after answer ("Is the answer correct? (Yes/No)"); and several open-source **reward models**, including AceMath-RM-7B (Liu et al., 2024), VersaPRM (Zeng et al., 2025), Math-Shepherd-Mistral-7B (Wang et al., 2023), and Qwen2.5-Math-7B-PRM800K / Qwen2.5-Math-PRM-7B (Zhang et al., 2025d). For GenRM, we apply the inference model itself to generate responses.

**Evaluation strategies** We consider three used inference-time aggregation strategies:

- **Self-consistency (a.k.a. majority voting)** (Wang et al., 2022): Generates multiple reasoning traces independently and selects the most frequent final answer by majority vote. We also consider its weighted variant, which assigns a weight $w(a_i)$ to each candidate answer $a_i$ based on its associated confidence score or verifier output. The final prediction is chosen as

$$a^* = \arg\max_{a \in \mathcal{A}} \sum_{i=1}^{N} w(a_i)\mathbb{I}[a_i = a], \tag{8}$$

  where $\mathcal{A}$ is the set of answers, and $\mathbb{I}[\cdot]$ is the indicator function.

- **Best-of-$N$ (BoN)**: Generates $N$ candidate solutions and selects the one with the highest score:

$$a^* = \arg\max_{i \in \{1,...,N\}} w(a_i). \tag{9}$$

  This method relies solely on model calibration and often suffers when calibration is poor.

- **Efficient BoN variants**: Inspired by Wang et al. (2025), we design pruning-based strategies to reduce computation while maintaining performance:
    - *DROP 10*: at every 10 tokens, discard the lowest-scoring trace until one remains.
    - *STOP 600*: at token 600, stop all other traces and continue with the highest-scoring one.
    - *HALF 300*: at every 300 tokens, remove the bottom half of traces until one remains.

In addition to the above baselines, we also report three standard metrics widely used in reasoning benchmarks: *Pass@1*, the average accuracy of a single trace; *Pass@k*, the probability that at least one correct solution appears among $k$ sampled traces (Chen et al., 2021); and *Maj@k*, the accuracy of unweighted majority voting. Notably, *Pass@k* serves as an upper bound for analysis.

It is worth noting that Self-Consistency and Best-of-$N$ incur the same computational cost, since both require completing all reasoning traces, though the former generally yields higher accuracy. In contrast, Best-of-$N$ can be adapted into more efficient variants by terminating most traces early and retaining only a single full trajectory, thereby reducing cost substantially. For instance, with 128 sampled traces, the *DROP 10*, *STOP 600*, and *HALF 300* strategies require only 81,280, 76,200, and 70,200 additional tokens respectively, along with 8,256, 128, and 252 verification calls. Assuming an average output length of 6,000 tokens, these approaches reduce computation by nearly 90% compared to standard Best-of-$N$. Since each verification call is equivalent to a one-token forward pass in our method, the overhead introduced by OTV is practically negligible.

## 4.2 MAIN RESULTS

In this section, we present the main evaluation results of OTV. Table 1 reports weighted majority voting accuracy against various baselines under the offline setting, where complete reasoning traces are required. Across all three model scales, OTV consistently delivers the best performance. We observe that while DeepConf provides improvements over majority voting in most cases, its gains remain limited. Methods that rely on the *verifier paradigm* or *reward models* also underperform compared to OTV, regardless of whether the verifier is the reasoning model itself (e.g., GenRM) or a well-trained external model. The key distinction is that OTV learns a *model-specific verifier* that directly captures the relationship between internal computation and confidence, enabling it to more accurately assess the quality of individual traces. This does not make the comparison unfair for two reasons. First, OTV only requires fine-tuning a small number of parameters with LoRA, making its training cost negligible, and its inference overhead is limited to a single one-token forward pass.

Table 1: **Weighted majority voting accuracy on AIME.** For PRMs, we follow prior work and take the score of the final token. For DeepConf and OTV, we follow Fu et al. (2025), computing the confidence as the mean over the last 100 and 2048 tokens, with the bottom 50% of traces filtered out. Each run samples 128 reasoning traces, and results are reported as averages over 64 runs. Pass@128 serves as an upper bound for parallel thinking since ground-truth is unavailable at inference.

| | | Qwen3-4B-Instruct-2507 | | Qwen3-8B | | DAPO-Qwen-32B | |
| --- | --- | --- | --- | --- | --- | --- | --- |
| | | AIME24 | AIME25 | AIME24 | AIME25 | AIME24 | AIME25 |
| | Pass@128 | 91.46 | 83.13 | 69.58 | 56.98 | 83.75 | 68.75 |
| *Internal* | Pass@1 | 60.29 | 46.67 | 26.29 | 19.32 | 51.77 | 36.42 |
| | Maj@128 | 75.42±1.61 | 66.46±1.16 | 44.22±2.11 | 28.44±2.06 | 66.72±0.93 | 41.77±1.76 |
| | DeepConf | 77.76±2.43 | 66.77±1.18 | 45.73±1.50 | 31.87±2.35 | 66.77±1.17 | 44.79±1.65 |
| | GenRM | 79.11±2.06 | 66.72±1.82 | 44.38±2.27 | 32.66±2.30 | 66.72±0.59 | 42.55±1.66 |
| *External* | AceMath-RM-7B | 67.66±1.74 | 60.00±2.36 | 44.90±2.70 | 26.72±2.18 | 61.15±2.22 | 42.76±2.00 |
| | VersaPRM-8B | 75.52±1.69 | 66.72±1.10 | 44.48±2.30 | 29.11±2.22 | 66.98±0.97 | 43.18±1.81 |
| | Math-Shepherd-7B | 74.27±1.50 | 66.56±1.02 | 44.22±1.79 | 26.04±1.94 | 65.73±2.24 | 43.65±1.93 |
| | Qwen2.5-PRM800K-7B | 75.16±1.66 | 66.35±1.14 | 44.22±2.30 | 29.32±2.57 | 67.03±1.46 | 48.18±3.48 |
| | Qwen2.5-PRM-7B | 78.18±1.85 | 63.49±2.39 | 45.10±2.35 | 29.48±2.63 | 67.45±2.26 | 46.77±3.06 |
| | OTV (Ours) | **83.33**±1.57 | **69.32**±1.46 | **46.56**±1.25 | **33.85**±1.69 | **70.83**±1.86 | **49.58**±1.10 |

Table 2: **Accuracy under BoN and three efficient variants with** $N = 128$. Numbers in parentheses denote the average length (tokens) of the final selected trace. Results are averaged over 64 runs.

| | Best-of-N@128 | | DROP 10@128 | | STOP 600@128 | | HALF 300@128 | |
| --- | --- | --- | --- | --- | --- | --- | --- | --- |
| | AIME24 | AIME25 | AIME24 | AIME25 | AIME24 | AIME25 | AIME24 | AIME25 |
| | *Qwen3-4B-Instruct-2507* | | | | | | | |
| VersaPRM-8B | 54.48 (6560) | 43.28 (6132) | 60.52 (2589) | 43.44 (6334) | 63.75 (3270) | 46.77 (6113) | 59.06 (2981) | 37.24 (6438) |
| Math-Shepherd-7B | **73.59** (5820) | **66.51** (5824) | 61.61 (5679) | 46.25 (6116) | 57.45 (5989) | 47.45 (6218) | 54.64 (5722) | 45.36 (6118) |
| Qwen2.5-PRM800K-7B | 69.90 (4891) | 45.10 (6202) | **66.77** (2196) | 42.86 (6445) | 60.31 (3434) | 49.53 (6409) | 62.24 (2919) | 45.21 (6619) |
| Qwen2.5-PRM-7B | 71.77 (3720) | 53.33 (3948) | 63.80 (3040) | 45.83 (6588) | 65.73 (4211) | 44.95 (6304) | 66.46 (3173) | 45.73 (6416) |
| DeepConf | 64.95 (9664) | 43.07 (9322) | 62.86 (7044) | 40.78 (7260) | 59.90 (7150) | 40.89 (7084) | 61.98 (6742) | 43.54 (6318) |
| OTV (Ours) | 73.44 (5447) | 53.91 (5416) | 63.39 (4427) | **46.46** (3225) | 63.75 (4431) | 49.11 (6542) | 67.03 (4132) | **49.02** (3170) |
| | *DAPO-Qwen-32B* | | | | | | | |
| VersaPRM-8B | 48.80 (5061) | 31.04 (4447) | **59.79** (5263) | 39.48 (4796) | 53.12 (5432) | 37.66 (5005) | 49.32 (5744) | 36.61 (5046) |
| Math-Shepherd-7B | 62.34 (5051) | 42.40 (4570) | 55.52 (4819) | 39.22 (4475) | 55.21 (5555) | 36.46 (5063) | 58.80 (4983) | 42.76 (4919) |
| Qwen2.5-PRM800K-7B | 54.17 (4722) | **47.81** (4426) | 53.28 (5585) | 41.20 (4689) | 49.48 (5260) | 40.89 (4523) | 47.40 (5732) | 40.00 (4528) |
| Qwen2.5-PRM-7B | 57.03 (4888) | 47.24 (4481) | 55.52 (5525) | 33.65 (5351) | 55.57 (4939) | 35.31 (5068) | 51.98 (5660) | 36.09 (4967) |
| DeepConf | 53.92 (5101) | 38.91 (3957) | 50.52 (6382) | 37.08 (4398) | 51.82 (7176) | 37.76 (4353) | 50.94 (3772) | 36.77 (4449) |
| OTV (Ours) | **63.18** (4623) | 47.08 (4079) | 55.95 (3397) | **50.68** (2926) | 53.54 (3211) | **48.23** (2577) | 55.05 (3436) | **46.98** (2991) |

Second, for widely available open-source models, our framework can be readily applied to train dedicated verifiers, which in practice makes OTV no less general than existing external approaches.

Table 2 compares OTV with Best-of-$N$ and its three efficient variants. For Best-of-$N$, all traces must be fully generated, which incurs the same computational cost as weighted majority voting. Thus, its performance should be considered alongside the weighted majority voting results in Table 1. We observe that, on average, Best-of-$N$ underperforms weighted majority voting by more than 10% in accuracy, reflecting the inherent inefficiency of relying solely on maximum-score selection. Although some reward models (e.g., Math-Shepherd-7B) perform competitively in this setting, OTV still achieves consistently strong results.

Under the efficient variants, OTV achieves the best or near-best accuracy across most configurations. Compared with Best-of-$N$, OTV typically selects shorter completions (about 20% fewer tokens), a property not observed in other methods. This behavior arises from our linearly increasing pseudo-confidence: shorter correct completions receive higher confidence than longer, less certain ones at the same token index. Among the variants, *HALF 300* offers the best trade-off between efficiency and accuracy, and OTV shows clear advantages in this setting. In summary, these results highlight that model-native verification is a reliable and efficient alternative to generic scoring methods.

### 4.3 VISUALIZATION

Figure 4 visualizes the confidence trajectories of different scoring methods on three representative AIME24 problems. Each curve corresponds to one reasoning trace, with red traces ending in correct

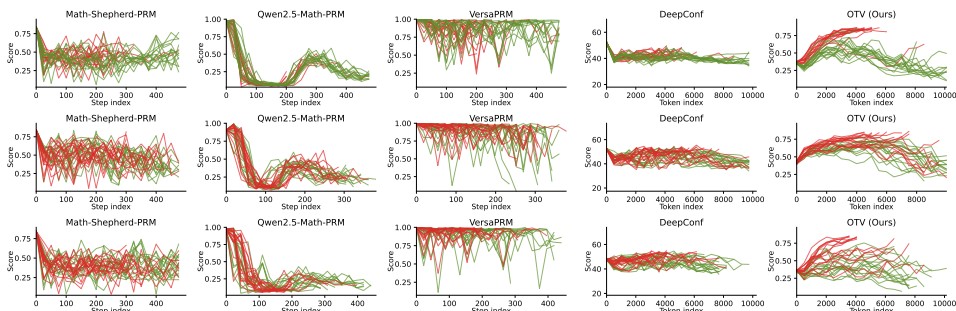

Figure 4: **Trace-level confidence dynamics on AIME24 problems #3, #9, and #22.** Each plot shows 64 sampled reasoning traces scored by three PRMs, DeepConf, and Qwen3-4B-Instruct-OTV (Ours). Red curves correspond to correct final answers and green to incorrect ones.

answers and green traces ending in incorrect ones. While existing PRMs and DeepConf often yield entangled trajectories with limited separation between correct and incorrect traces, OTV produces a much clearer distinction: correct traces steadily accumulate higher confidence, whereas incorrect ones remain suppressed. This separability demonstrates that OTV more faithfully captures the underlying reasoning quality, enabling reliable discrimination across diverse problem instances.

In addition to the case-level trajectories shown above and in Appendix D, we also present fine-grained token-level visualizations in Appendix B. These plots overlay the reasoning text with token-level confidence predictions, illustrating how OTV dynamically separates promising reasoning paths from misleading ones. A common pattern is that confidence rises sharply after key computational steps. While these examples are too lengthy to include in the main text, they further demonstrate the interpretability of OTV at the granularity of individual reasoning steps.

## 5 ANALYSIS

### 5.1 ABLATION STUDY

We conduct ablation studies to better understand the design choices in OTV. Specifically, we examine (i) the impact of LoRA rank on performance, (ii) different heuristic schemes for confidence, and (iii) the impact of trace filtering in weighted majority voting, following Fu et al. (2025).

**Effect of LoRA rank.** We first examine the impact of LoRA rank on OTV's performance. Using the *Qwen3-4B-Instruct* model with weighted majority voting, Figure 5 reports training loss and accuracy as reasoning traces scale from 4 to 128. Results show that higher ranks consistently reduce training loss and improve accuracy. Compared to the *probe* baseline, OTV achieves clear performance gains, confirming the effectiveness of OTV and indicating that leveraging the full KV cache provides clearer benefits than approaches restricted to final hidden states (Zhang et al., 2025a). Notably, even moderate ranks (e.g., $r = 16$) provide substantial improvements.

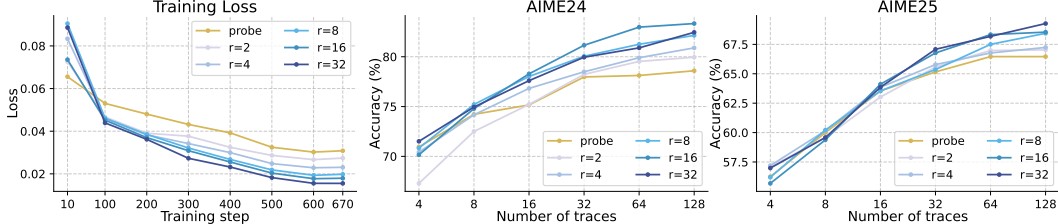

Figure 5: Training loss (left) and weighted majority voting accuracy on AIME24 (middle) and AIME25 (right). Results are averaged over 64 runs while scaling with the number of reasoning traces. The *probe* baseline trains only the regression head without LoRA or KV cache reuse.

**Pseudo-confidence design.** We next study the impact of different heuristic schemes for confidence supervision. As shown in Table 3, the *Constant* biases toward shorter traces but slightly reduces accuracy, whereas the *Linear*, *Step*, and *Wave* schemes adhere to our principle of monotonically increasing confidence and deliver comparable performance with balanced trace lengths.

Table 3: Comparison of different heuristic schemes for confidence of *Qwen3-4B-Instruct-OTV*.

| | Majority Voting@32 | | Majority Voting@128 | | Best-of-N@128 | | HALF 300@128 | |
|---|---|---|---|---|---|---|---|---|
| | AIME24 | AIME25 | AIME24 | AIME25 | AIME24 | AIME25 | AIME24 | AIME25 |
| Linear | **81.15**±3.13 | 66.93±3.60 | **83.33**±1.57 | **69.32**±1.46 | 73.44 (5447) | 53.91 (5416) | 67.03 (4132) | **49.02** (3170) |
| Constant | 80.21±2.56 | 68.49±2.04 | 81.41±1.65 | 64.37±4.28 | 72.76 (**4340**) | 54.27 (**4362**) | 66.41 (**3951**) | 49.27 (**4503**) |
| Step | 80.99±2.81 | **68.80**±1.60 | 82.40±1.50 | 66.20±3.00 | **77.29** (5351) | **55.36** (5452) | 70.63 (4231) | 37.60 (7095) |
| Wave | 80.36±2.95 | 67.92±1.61 | 81.87±1.65 | 66.25±3.70 | 72.60 (5226) | 54.90 (5152) | **71.30** (4134) | 48.33 (5736) |

Table 4: Accuracy of OTV with weighted majority voting under different filtering schemes.

| | Maximum | | | | Mean | | | | Minimum | | | |
|---|---|---|---|---|---|---|---|---|---|---|---|---|
| | $\eta = 0$ | $\eta = 25\%$ | $\eta = 50\%$ | $\eta = 75\%$ | $\eta = 0$ | $\eta = 25\%$ | $\eta = 50\%$ | $\eta = 75\%$ | $\eta = 0$ | $\eta = 25\%$ | $\eta = 50\%$ | $\eta = 75\%$ |
| Last 100 | 78.39 | 80.21 | 82.34 | 81.82 | 82.08 | 82.97 | 83.33 | 82.76 | 82.86 | 82.60 | 82.76 | **83.96** |
| Last 400 | 77.50 | 80.36 | 82.66 | 82.86 | 81.93 | 82.81 | 83.33 | 83.23 | 82.86 | 83.23 | 83.28 | 83.75 |
| Last 1600 | 75.78 | 78.65 | 81.25 | 82.14 | 80.99 | 82.71 | 83.23 | 82.24 | 83.23 | 83.23 | 82.97 | 79.95 |
| Last 10% | 76.77 | 80.21 | 82.76 | 81.46 | 82.19 | 83.02 | 83.28 | 82.19 | 83.07 | 83.07 | 83.12 | 82.29 |
| Last 100% | 75.78 | 76.82 | 80.42 | 80.36 | 75.94 | 79.53 | 82.19 | 80.00 | 80.10 | 80.83 | 81.25 | 79.17 |

**Trace filtering.** Inspired by DeepConf (Fu et al., 2025), we evaluate *Qwen3-4B-Instruct-OTV* on AIME24 with weighted majority voting under different filtering schemes. Confidences are aggregated over the last $N$ tokens and the lowest $\eta$ fraction of traces is removed before voting. Table 4 shows that averaging or minimizing over the last 100 tokens with filtering yields the best results, indicating that confidence estimates derived from tokens near the end of reasoning are more reliable.

## 5.2 Evaluation on the Pre-trained Base Model

For further analysis, we evaluate the pre-trained base model, *Qwen3-4B-Base*, which differs from the post-trained models used in Section 4. Here, we investigate whether OTV can improve its raw mathematical reasoning performance. OTV is calibrated on the widely used MetaMathQA dataset (Yu et al., 2023b) and evaluated on GSM8K (Cobbe et al., 2021), a benchmark of grade-school math problems. This experiment requires only a single 96GB GPU for supervised fine-tuning.

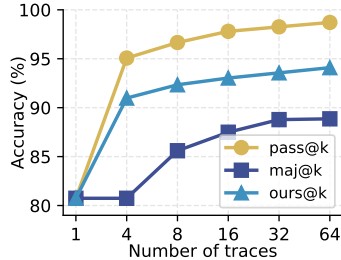

Figure 6: Accuracy on GSM8K.

As shown in Figure 6, OTV substantially improves performance over *Maj@k*, narrowing the gap toward *Pass@k*. Beyond accuracy, we further examine robustness by perturbing reasoning traces with both benign edits and adversarial errors. OTV demonstrates strong stability under harmless modifications (e.g., paraphrasing or formatting changes) while sharply decreasing confidence for adversarially corrupted traces, confirming its ability to distinguish between valid and flawed reasoning (see Appendix C for detailed examples).

## 6 Conclusion and Future Work

In this paper, we proposed *One-Token Verification* (OTV), a lightweight framework for reasoning quality assessment that augments LLMs with a LoRA-based verifier operating directly on the KV cache. OTV enables anytime, token-level confidence estimation with minimal overhead. Extensive experiments on multiple reasoning LLMs and math benchmarks demonstrate that OTV consistently outperforms prior baselines in parallel thinking, while also favoring shorter and more accurate reasoning traces under efficient settings.

Looking forward, an important direction is to explore tighter integration between the base model and verifier. Since the verifier can be viewed as a lightweight hyper-network, future work could investigate joint or continual updates, ensuring that the verifier adapts as the base model evolves. Another promising work is to extend the binary (correct/incorrect) confidence into a ternary setup, introducing an "unknown" state that allows the verifier to abstain when uncertain. This would enable selective prediction and open up opportunities for active learning and verifier-guided reinforcement learning, where uncertain samples are prioritized for further training.

ETHICS STATEMENT

This paper presents work whose goal is to advance the field of Machine Learning. The authors have read and comply with the ICLR Code of Ethics. The research did not involve human subjects, animal experiments, or personally identifiable data. All experiments were conducted on publicly available benchmarks and open-source models. We have carefully considered the broader impacts and believe that this work poses no foreseeable risks of harm, while contributing to the development of trustworthy and efficient reasoning systems.

REPRODUCIBILITY STATEMENT

The authors have made significant efforts to ensure the reproducibility of results. Section 4.1 details the experimental setup, including datasets, model configurations, and hyperparameter settings. Additional ablations in Section 5.1 further analyze the effect of training choices. Algorithmic details are included in Appendix E. We will release our code and scripts for training and evaluation, along with the fine-tuned verifier models (e.g., Qwen3-4B-Instruct-OTV), to facilitate reuse and comparison within the community.

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

# A  LLM USAGE STATEMENT

In the preparation of this manuscript, LLM was utilized solely as a language-editing tool. Its application was restricted to improving clarity, and conciseness, and correcting grammatical errors. The LLM was not used for generating or analyzing any of the substantive content, ideas, or results.

# B  FINE-GRAINED VISUALIZATION OF CONFIDENCE PREDICTIONS

To complement the main results, we provide token-level visualizations of OTV's confidence predictions, with reasoning text color-coded from green (lowest confidence) to red (highest confidence). Six examples highlight three comparative settings: (i) correct vs. incorrect traces for the same problem (Examples 1–2), where confidence clearly distinguishes correctness; (ii) a shorter answer vs. a longer answer (Examples 3–4), showing that shorter traces yield consistently higher confidence while longer ones exhibit fluctuations, reflecting intermediate uncertainty; and (iii) correct vs. incorrect traces from a larger model on the same problem (Examples 5–6), where confidence declines toward the end for both, but overall scores remain much higher for the correct traces.

---

**Illustrative Example 1: AIME'25, Problem 23, Qwen3-4B-Instruct-OTV**

There are $n$ values of $x$ in the interval $0 < x < 2\pi$ where $f(x) = \sin(7\pi \cdot \sin(5x)) = 0$. For $t$ of these $n$ values of $x$, the graph of $y = f(x)$ is tangent to the x-axis. Find $n + t$. (Correct Answer: 149)

We are given the function: $f(x) = \sin(7\pi \cdot \sin(5x))$ We are to find: - $n$: the number of values of $x$ in the interval $0 x 2\pi$ where $f(x) = 0$ - $t$: the number of those $n$ values where the graph is **tangent** to the x-axis (i.e., where $f(x) = 0$ and $f'(x) = 0$) - Then compute $n + t$ — Step 1: When is $f(x) = 0$? We have: $f(x) = \sin(7\pi \cdot \sin(5x)) = 0$ The sine function is zero when its argument is an integer multiple of $\pi$: $7\pi \cdot \sin(5x) = k\pi$ for some integer $k$ Divide both sides by $\pi$: $7\sin(5x) = k \Rightarrow \sin(5x) = \frac{k}{7}$ Now, since $\sin(5x) \in [-1, 1]$, we require: $-1 \le \frac{k}{7} \le 1 \Rightarrow -7 \le k \le 7$ So $k = -7, -6, \ldots, 0, \ldots, 6, 7$ That's 15 integer values of $k$. For each such $k$, we solve: $\sin(5x) = \frac{k}{7}$ We now want to count how many solutions $x \in (0, 2\pi)$ satisfy this. — Step 2: Count total number of solutions $n$ Let's consider $\sin(5x) = c$, where $c = \frac{k}{7}$, and $c \in [-1, 1]$. For each $c \in (-1, 1)$, the equation $\sin(5x) = c$ has **two** solutions per period of $5x$, i.e., for every interval of length $\frac{2\pi}{5}$ in $x$, there are two solutions (since sine is periodic and symmetric). But we need to count how many solutions exist in $x \in (0, 2\pi)$. Let's define $\theta = 5x$. Then as $x$ goes from 0 to $2\pi$, $\theta$ goes from 0 to $10\pi$. So $\theta \in (0, 10\pi)$ We now solve $\sin(\theta) = \frac{k}{7}$ for $\theta \in (0, 10\pi)$ For each $k$, we consider how many solutions $\theta \in (0, 10\pi)$ satisfy $\sin(\theta) = \frac{k}{7}$ — General behavior of $\sin(\theta) = c$: - If $c = \pm 1$, there is **one** solution per period of $2\pi$, but only at the peaks/troughs. - If $c \in (-1, 1)$, there are **two** solutions per $2\pi$ interval (one in $(0, \pi)$, one in $(\pi, 2\pi)$) But note: $\theta \in (0, 10\pi)$, which is **5 full periods** of $2\pi$ So total length: $10\pi$, so 5 periods. — Let's consider each $k$: - For $k = \pm 7$: $\sin(5x) = \pm 1$ - $\sin(\theta) = 1$: occurs at $\theta = \frac{\pi}{2} + 2\pi m$ - $\sin(\theta) = -1$: occurs at $\theta = \frac{3\pi}{2} + 2\pi m$ In $(0, 10\pi)$, how many such values? - $\theta = \frac{\pi}{2}, \frac{5\pi}{2}, \frac{9\pi}{2}, \frac{13\pi}{2}, \frac{17\pi}{2}, \frac{21\pi}{2}, \frac{25\pi}{2}$ $\frac{25\pi}{2} = 12.5\pi 10\pi$? $10\pi = 20\pi/2$, so up to $m$ such that $\frac{\pi}{2} + 2\pi m 10\pi$ $\frac{\pi}{2} + 2\pi m 10\pi$ $2\pi m 10\pi - \frac{\pi}{2} = \frac{19\pi}{2}$ $m \frac{19}{4} = 4.75$ So $m = 0, 1, 2, 3, 4 \rightarrow$ 5 values So $\sin(\theta) = 1$ has 5 solutions Similarly, $\sin(\theta) = -1$: $\theta = \frac{3\pi}{2} + 2\pi m$ $\frac{3\pi}{2} + 2\pi m 10\pi$ $2\pi m 10\pi - \frac{3\pi}{2} = \frac{17\pi}{2}$ $m 17/4 = 4.25 \rightarrow m = 0, 1, 2, 3, 4$ $\rightarrow$ 5 values So total for $k = \pm 7$: $5 + 5 = 10$ solutions - For $k = 0$: $\sin(\theta) = 0$ Solutions at $\theta = \pi m$, $m = 1, 2, 3, 4, 5, 6, 7, 8, 9$ $\rightarrow$ since $\theta \in (0, 10\pi)$ $\theta = \pi, 2\pi, 3\pi, 4\pi, 5\pi, 6\pi, 7\pi, 8\pi, 9\pi \rightarrow m = 1$ to 9 $m = 10 \rightarrow 10\pi$, which is excluded since $\theta 10\pi$ So 9 solutions - For $k = \pm 1, \pm 2, \pm 3, \pm 4, \pm 5, \pm 6$: these are all in $(-1, 1)$, and not $\pm 1$ For each such $c \in (-1, 1)$, in each interval of $2\pi$, there are **two** solutions. Since $\theta \in (0, 10\pi)$, which is 5 full periods, each such $c$ gives $2 \times 5 = 10$ solutions. How many such $k$? $k = \pm 1, \pm 2, \pm 3, \pm 4, \pm 5, \pm 6 = 12$ values So total solutions from these: $12 \times 10 = 120$ Wait — but we must be cautious: are these values distinct? Actually, each $k$ gives a different equation $\sin(5x) = k/7$, and each has its own solutions. So yes, we are counting solutions for each $k$, so we can sum over all $k$. So total $n$: - $k = \pm 7$: 10 solutions - $k = 0$: 9 solutions - $k = \pm 1, \pm 2, \pm 3, \pm 4, \pm 5, \pm 6$: 12 values, each with 10 solutions $\rightarrow 12 \times 10 = 120$ Total: $n = 10 + 9 + 120 = 139$ Wait — is that correct? Wait: $k = -7$ to 7: total 15 values. We broke it as: - $k = \pm 7$: 2 values $\rightarrow$ 10 solutions - $k = 0$: 1 value $\rightarrow$ 9 solutions - $k = \pm 1$ to $\pm 6$: 12 values $\rightarrow$ 120 solutions Total: $10 + 9 + 120 = 139$ But let's double-check the number of solutions for $\sin(\theta) = c$ in $(0, 10\pi)$ For $c \in (-1, 1)$, $\sin(\theta) = c$ has two solutions per $2\pi$ interval. In $(0, 10\pi)$, which is 5 full periods, there are $5 \times 2 = 10$ solutions per $c$. Yes. For $c = \pm 1$, only one solution per $2\pi$ interval. In $(0, 10\pi)$, 5 periods $\rightarrow$ 5 solutions per $\pm 1$ So $\sin(\theta) = 1$: 5 solutions $\sin(\theta) = -1$: 5 solutions $\rightarrow$ total 10 For $\sin(\theta) = 0$: solutions at $\theta = \pi, 2\pi, 3\pi, 4\pi, 5\pi, 6\pi, 7\pi, 8\pi, 9\pi$ $\theta = 10\pi$ is excluded $\rightarrow$ so 9 solutions Yes. So total $n = 139$ But wait — are all these $x$ values in $(0, 2\pi)$? Yes, because $\theta = 5x \in (0, 10\pi)$, so $x \in (0, 2\pi)$ So yes. So $n = 139$ — Step 3: Find $t$: number of $x$ values where $f(x) = 0$ **and** $f'(x) = 0$ We want the graph to be tangent to the x-axis — that is, a **double root** — so $f(x) = 0$ and $f'(x) = 0$ We need to compute $f'(x)$ and find where both $f(x) = 0$ and $f'(x) = 0$ Let: $f(x) = \sin(7\pi \cdot \sin(5x))$ Let $u = 7\pi \cdot \sin(5x)$, so $f(x) = \sin(u)$ Then: $f'(x) = \cos(u) \cdot \frac{d}{dx}(7\pi \sin(5x)) = \cos(7\pi \sin(5x)) \cdot 7\pi \cdot 5\cos(5x)$ $f'(x) = 35\pi \cos(7\pi \sin(5x)) \cdot \cos(5x)$ We want $f(x) = 0$ **and** $f'(x) = 0$ So: 1. $\sin(7\pi \sin(5x)) = 0$ 2. $\cos(7\pi \sin(5x)) \cdot \cos(5x) = 0$ From (1): $7\pi \sin(5x) = k\pi \Rightarrow \sin(5x) = \frac{k}{7}$ for integer $k \in [-7, 7]$ From (2): $\cos(7\pi \sin(5x)) \cdot \cos(5x) = 0$ So either: - $\cos(7\pi \sin(5x)) = 0$, or - $\cos(5x) = 0$ But note: $\sin(5x) = \frac{k}{7}$, so $7\pi \sin(5x) = k\pi$ So $\cos(7\pi \sin(5x)) = \cos(k\pi)$ Because $7\pi \sin(5x) = k\pi$ so $\cos(k\pi) = (-1)^k$ So $\cos(7\pi \sin(5x)) = (-1)^k$ This is **never zero** unless $(-1)^k = 0$, which is impossible. So $\cos(7\pi \sin(5x)) \ne 0$ for any integer $k$ Therefore, the only way for $f'(x) = 0$ is if $\cos(5x) = 0$ So we require: - $\sin(5x) = \frac{k}{7}$ - $\cos(5x) = 0$ Now, when is $\cos(5x) = 0$? That happens when $5x = \frac{\pi}{2} + m\pi$, for integer $m$ So $x = \frac{\pi}{10} + \frac{m\pi}{5}$ We now need to find values of $x \in (0, 2\pi)$ such that: - $\cos(5x) = 0$ - $\sin(5x) = \frac{k}{7}$ for some integer $k$ But when $\cos(5x) = 0$, $\sin(5x) = \pm 1$ So $\sin(5x) = \pm 1$ Therefore, $\frac{k}{7} = \pm 1 \Rightarrow k = \pm 7$ So only when $k = \pm 7$, we can have $\cos(5x) = 0$ and $\sin(5x) = \pm 1$ Therefore, the only possible tangent points occur when $\sin(5x) = \pm 1$ Now, we already found that when $\sin(5x) = \pm 1$, $f(x) = \sin(7\pi \cdot (\pm 1)) = \sin(\pm 7\pi) = 0$ Because $\sin(7\pi) = \sin(\pi) = 0$ So $f(x) = 0$ And we just showed $f'(x) = 0$ when $\cos(5x) = 0$ Therefore, at all points where $\sin(5x) = \pm 1$, we have both $f(x) = 0$ and $f'(x) = 0$ Therefore, **tangent points occur precisely when $\sin(5x) = \pm 1$** We already computed that in $(0, 10\pi)$, $\theta = 5x$, $\sin(\theta) = 1$ has 5 solutions, $\sin(\theta) = -1$ has 5 solutions $\rightarrow$ total 10 solutions So $t = 10$ But wait — are these points distinct? Yes — for $\sin(\theta) = 1$: $\theta = \frac{\pi}{2}, \frac{5\pi}{2}, \frac{9\pi}{2}, \frac{13\pi}{2}, \frac{17\pi}{2}$ $\frac{17\pi}{2} = 8.5\pi 10\pi$, next would be $\frac{21\pi}{2} = 10.5\pi 10\pi$ Similarly, $\sin(\theta) = -1$: $\theta = \frac{3\pi}{2}, \frac{7\pi}{2}, \frac{11\pi}{2}, \frac{15\pi}{2}, \frac{19\pi}{2}$ $\frac{19\pi}{2} = 9.5\pi 10\pi$, next $\frac{23\pi}{2} = 11.5\pi 10\pi$ All are distinct and in $(0, 10\pi)$ So total $t = 5 + 5 = 10$ — Final Step: Compute $n + t$ We found: - $n = 139$ - $t = 10$ So: $n + t = 139 + 10 = 149$ Answer: 149

---

### Illustrative Example 2: AIME'25, Problem 23, Qwen3-4B-Instruct-OTV

There are $n$ values of $x$ in the interval $0 < x < 2\pi$ where $f(x) = \sin\left(7\pi \cdot \sin(5x)\right) = 0$. For $t$ of these $n$ values of $x$, the graph of $y = f(x)$ is tangent to the $x$-axis. Find $n + t$. (Correct Answer: 149)

We are given the function: $f(x) = \sin\left(7\pi \cdot \sin(5x)\right)$ We are to find: - The total number $n$ of values of $x$ in the interval $0 x 2\pi$ such that $f(x) = 0$ - Among these $n$ values, how many $t$ are such that the graph is **tangent** to the $x$-axis (i.e., $f(x) = 0$ and $f'(x) = 0$) - Then compute $n + t$ — Step 1: When is $f(x) = 0$? We have: $f(x) = \sin\left(7\pi \cdot \sin(5x)\right) = 0$ The sine function is zero when its argument is an integer multiple of $\pi$: $7\pi \cdot \sin(5x) = k\pi$ for some integer $k$ Divide both sides by $\pi$: $7\sin(5x) = k \Rightarrow \sin(5x) = \frac{k}{7}$ Now, since $\sin(5x) \in [-1, 1]$, we must have: $-1 \le \frac{k}{7} \le 1 \Rightarrow -7 \le k \le 7$ So $k$ ranges from $-7$ to $7$, inclusive. That's 15 integer values. For each such $k$, we solve: $\sin(5x) = \frac{k}{7}$ We want to find how many **solutions** $x \in (0, 2\pi)$ satisfy this equation. Let's define $\theta = 5x$. Then $x \in (0, 2\pi) \Rightarrow \theta \in (0, 10\pi)$ So we are solving $\sin(\theta) = \frac{k}{7}$ for $\theta \in (0, 10\pi)$ — Step 2: Number of solutions to $\sin(\theta) = c$ in $(0, 10\pi)$ For a fixed $c \in (-1, 1)$, the equation $\sin(\theta) = c$ has **two solutions per period** of $2\pi$, except when $c = \pm 1$, where it has one solution per period (at the peak/trough). So we analyze based on the value of $k$: Case 1: $k = 0$ Then $\sin(5x) = 0$ Solutions: $\sin(\theta) = 0$, $\theta \in (0, 10\pi)$ Solutions occur at $\theta = \pi, 2\pi, 3\pi, 4\pi, 5\pi, 6\pi, 7\pi, 8\pi, 9\pi$ Note: $\theta = 0$ and $10\pi$ are excluded because $x = 0$ and $x = 2\pi$ are excluded. So $\theta = \pi, 2\pi, 3\pi, 4\pi, 5\pi, 6\pi, 7\pi, 8\pi, 9\pi \to 9$ solutions Each corresponds to $x = \theta/5$, so $x = \pi/5, 2\pi/5, \ldots, 9\pi/5$ All are in $(0, 2\pi)$ So **9 solutions** for $k = 0$ — Case 2: $k \ne 0$ We consider $k = \pm1, \pm2, \ldots, \pm7$ For each such $k$, $\frac{k}{7} \in (-1, 1)$, so $\sin(\theta) = \frac{k}{7}$ has **two solutions per period** of $2\pi$ in $(0, 10\pi)$ How many periods in $(0, 10\pi)$? $\frac{10\pi}{2\pi} = 5$ full periods So in each period, two solutions $\to$ total of $5 \times 2 = 10$ solutions per value of $k \in \{-7, \ldots, 7\} \setminus \{0\}$ But wait — are there any exceptions? Only when $\sin(\theta) = \pm1$, i.e., $k = \pm7$ Let's check: - $k = 7$: $\sin(5x) = 1$ - $k = -7$: $\sin(5x) = -1$ For these, $\sin(\theta) = \pm1$ In each $2\pi$ interval, $\sin(\theta) = 1$ occurs once: at $\theta = \frac{\pi}{2} + 2\pi m$ Similarly, $\sin(\theta) = -1$ at $\theta = \frac{3\pi}{2} + 2\pi m$ So in $(0, 10\pi)$, how many such solutions? - For $\sin(\theta) = 1$: $\theta = \frac{\pi}{2}, \frac{5\pi}{2}, \frac{9\pi}{2}, \frac{13\pi}{2}, \frac{17\pi}{2}$ Check: $\frac{\pi}{2} + 2\pi m 10\pi$ $\frac{\pi}{2} + 2\pi m 10\pi$ Divide by $\pi$: $\frac{1}{2} + 2m 10 \Rightarrow 2m 9.5 \Rightarrow m 4.75$ So $m = 0, 1, 2, 3, 4 \to 5$ values Similarly for $\sin(\theta) = -1$: $\theta = \frac{3\pi}{2} + 2\pi m$ $\frac{3}{2} + 2m 10 \Rightarrow 2m 9.5 \Rightarrow m 4.75 \to m = 0, 1, 2, 3, 4 \to 5$ values So for $k = \pm7$, we have **5 solutions each** For other $k = \pm1, \pm2, \pm3, \pm4, \pm5, \pm6$, we have $\sin(\theta) = \frac{k}{7} \in (-1, 1)$, so **two solutions per period**, 5 periods $\to$ $5 \times 2 = 10$ solutions each — Step 3: Count total number of solutions $n$ List all $k$ from $-7$ to 7: - $k = 0$: 9 solutions - $k = \pm1, \pm2, \pm3, \pm4, \pm5, \pm6$: 6 values of $k$, each with 10 solutions $\to 6 \times 10 = 60$ - $k = \pm7$: each with 5 solutions $\to 2 \times 5 = 10$ Total $n = 9 + 60 + 10 = 79$ So $n = 79$ — Step 4: Find $t$ — number of values where the graph is **tangent** to the $x$-axis This means $f(x) = 0$ **and** $f'(x) = 0$ We need to find the values of $x$ where $f(x) = 0$ and $f'(x) = 0$ First, recall: $f(x) = \sin\left(7\pi \cdot \sin(5x)\right)$ Let $u(x) = 7\pi \cdot \sin(5x)$, so $f(x) = \sin(u(x))$ Then: $f'(x) = \cos(u(x)) \cdot u'(x)$ Compute $u'(x)$: $u'(x) = 7\pi \cdot \cos(5x) \cdot 5 = 35\pi \cos(5x)$ So: $f'(x) = \cos(7\pi \sin(5x)) \cdot 35\pi \cos(5x)$ We want $f'(x) = 0$ **and** $f(x) = 0$ So two conditions: 1. $\sin(7\pi \sin(5x)) = 0$ 2. $\cos(7\pi \sin(5x)) \cdot 35\pi \cos(5x) = 0$ Note that $35\pi \ne 0$, so we can divide: So condition 2 becomes: $\cos(7\pi \sin(5x)) = 0$ or $\cos(5x) = 0$ But from condition 1: $\sin(7\pi \sin(5x)) = 0$ Let $s = \sin(5x)$. Then: - $\sin(7\pi s) = 0$ - Either $\cos(7\pi s) = 0$ or $\cos(5x) = 0$ We are to find values where **both** conditions hold. So we consider two cases: — Case A: $\cos(5x) = 0$ Then $5x = \frac{\pi}{2} + m\pi \Rightarrow x = \frac{\pi}{10} + \frac{m\pi}{5}$ We need $x \in (0, 2\pi)$ So $0 \frac{\pi}{10} + \frac{m\pi}{5} 2\pi$ Divide by $\pi$: $0 \frac{1}{10} + \frac{m}{5} 2$ Multiply by 10: $0 1 + 2m 20$ $2m - 1 \Rightarrow m - 0.5 \to m \ge 0$ $2m 19 \Rightarrow m 9.5 \to m \le 9$ So $m = 0, 1, 2, \ldots, 9 \to 10$ values Now, for each such $x$, we must check whether $f(x) = 0$, i.e., $\sin(7\pi \sin(5x)) = 0$ At such $x$, $\cos(5x) = 0$, so $5x = \frac{\pi}{2} + m\pi$ Then $\sin(5x) = \sin\left(\frac{\pi}{2} + m\pi\right) = (-1)^m$ So $\sin(5x) = (-1)^m \in \{-1, 1\}$ Then $7\pi \sin(5x) = \pm7\pi$ So $\sin(7\pi \sin(5x)) = \sin(\pm7\pi) = \sin(7\pi) = 0$ Because $\sin(7\pi) = \sin(\pi) = 0$ So yes, $f(x) = 0$ Also, $f'(x) = 0$ because $\cos(5x) = 0$ So **all 10 values** of $x$ where $\cos(5x) = 0$ satisfy both $f(x) = 0$ and $f'(x) = 0$ So **t includes these 10 values** — Case B: $\cos(7\pi \sin(5x)) = 0$ **and** $\sin(7\pi \sin(5x)) = 0$ Is this possible? Let $\theta = 7\pi \sin(5x)$ We want: - $\sin(\theta) = 0$ - $\cos(\theta) = 0$ But $\sin(\theta) = 0$ and $\cos(\theta) = 0$ is **impossible** for any real $\theta$ Because $\sin^2(\theta) + \cos^2(\theta) = 1$ So both cannot be zero. Therefore, **no solutions** in this case. So the only points where $f(x) = 0$ and $f'(x) = 0$ are when $\cos(5x) = 0$ Thus, $t = 10$ — Final Step: Compute $n + t$ We found: - $n = 79$ - $t = 10$ So: Answer: 89

### Illustrative Example 3: AIME'24, Problem 17, Qwen3-4B-Instruct-OTV

There exist real numbers $x$ and $y$, both greater than 1, such that $\log_x\left(y^x\right) = \log_y\left(x^{4y}\right) = 10$. Find $xy$. (Correct Answer: 25)

We are given that: $\log_x(y^x) = 10$ and $\log_y(x^{4y}) = 10$ We are to find $xy$, where $x 1$, $y 1$. — Step 1: Simplify the first equation $\log_x(y^x) = 10$ Use logarithmic identity: $\log_b(a^c) = c \log_b a$ $\log_x(y^x) = x \log_x y = 10$ So: $x \log_x y = 10$ (Equation 1) — Step 2: Simplify the second equation $\log_y(x^{4y}) = 10$ Again, use $\log_b(a^c) = c \log_b a$: $4y \log_y x = 10$ (Equation 2) — Step 3: Use substitution Let's define $a = \log_x y$. Then from Equation 1: $x \cdot a = 10 \Rightarrow a = \frac{10}{x}$ Now, recall that $\log_x y = a \Rightarrow \log_y x = \frac{1}{a}$ (since logarithms are reciprocals) So from Equation 2: $4y \cdot \log_y x = 10 \Rightarrow 4y \cdot \frac{1}{a} = 10$ Substitute $a = \frac{10}{x}$: $4y \cdot \frac{1}{\frac{10}{x}} = 10 \Rightarrow 4y \cdot \frac{x}{10} = 10$ Simplify: $\frac{4xy}{10} = 10 \Rightarrow \frac{2xy}{5} = 10$ Multiply both sides by 5: $2xy = 50 \Rightarrow xy = 25$ — Answer: 25

---

### Illustrative Example 4: AIME'24, Problem 22, Qwen3-4B-Instruct-OTV

Let $\triangle ABC$ have circumcenter $O$ and incenter $I$ with $\overline{IA} \perp \overline{OI}$, circumradius 13, and inradius 6. Find $AB \cdot AC$. (Correct Answer: 468)

- - - - - - - - - - - - - - - - - - - - - - - - - - - - - - - - - - - - - - - - - - -

We are given the following about triangle $\triangle ABC$: - $O$ is the circumcenter. - $I$ is the incenter. - $IA \perp OI$. - Circumradius $R = 13$. - Inradius $r = 6$. - We are to find $AB \cdot AC$. — Step 1: Use known geometric identities We are given a perpendicularity condition: $IA \perp OI$. This is a strong geometric condition and suggests we can use known formulas or coordinate geometry. We aim to compute $AB \cdot AC$, which is a product of two sides. Let's recall some known identities and formulas: — Step 2: Use known formula involving $IA$, $OI$, and perpendicularity There is a known identity involving the distance between incenter $I$ and circumcenter $O$: $OI^2 = R(R - 2r)$ This is **Euler's formula** for the distance between incenter and circumcenter. Wait — actually, **no**, Euler's formula is: $OI^2 = R^2 - 2Rr$ Yes — the correct formula is: $OI^2 = R^2 - 2Rr$ Let's verify: - $R = 13$, $r = 6$ - So $OI^2 = 13^2 - 2 \cdot 13 \cdot 6 = 169 - 156 = 13$ So $OI^2 = 13$ — Step 3: Use the given perpendicularity $IA \perp OI$ We are told $IA \perp OI$. This is a key condition. Let's use vector geometry or coordinate geometry to exploit this. Let's place the triangle in the coordinate plane with convenient positions. — Step 4: Use coordinate geometry Let us place point $A$ at the origin, and use symmetry. But perhaps a better idea is to use known vector identities. There is a known vector identity: In triangle $ABC$, with circumcenter $O$, incenter $I$, and a point $A$, the condition $\vec{IA} \cdot \vec{OI} = 0$ (since $IA \perp OI$) implies: $\vec{IA} \cdot \vec{OI} = 0$ Let's define vectors with respect to point $A$. Alternatively, use known formula for $IA^2$: $IA^2 = \frac{bc}{(b+c)^2}\left((b+c)^2 - a^2\right)$ But that might be messy. There is a known formula: $IA^2 = \frac{bc}{(b+c)^2}(b+c-a)(b+c+a)$ Alternatively, standard formula: $IA^2 = r^2 + (s-a)^2$ Yes! That's a standard identity: In any triangle, the distance from the incenter $I$ to vertex $A$ is: $IA^2 = r^2 + (s-a)^2$ where $s$ is the semi-perimeter, and $a = BC$, $b = AC$, $c = AB$. We are to find $AB \cdot AC = bc$. So let's define: - $a = BC$ - $b = AC$ - $c = AB$ - $s = \frac{a+b+c}{2}$ - $r = 6$ - $R = 13$ - $OI^2 = 13$ We also have the identity: $IA^2 = r^2 + (s-a)^2$ We are told $IA \perp OI$, so the vectors $\vec{IA}$ and $\vec{OI}$ are perpendicular. We can use a known **vector identity** or **formula** involving this perpendicularity. — Step 5: Use known identity for perpendicularity of $IA$ and $OI$ There is a known result in triangle geometry: In triangle $ABC$, if $IA \perp OI$, then: $bc = \frac{r^2 + 4Rr}{\sin^2 A}$ Wait — not sure. Alternatively, a known result from olympiad geometry: If $IA \perp OI$, then: $\frac{r}{R} = \frac{1}{2}\left(1 - \frac{a^2}{b^2+c^2}\right)$ Not sure. Alternatively, we can use a known formula involving $OI^2$, $IA^2$, and angle between them. Since $IA \perp OI$, then the angle between vectors $\vec{IA}$ and $\vec{OI}$ is $90°$, so their dot product is zero. Let's use vector identities. Let's place point $A$ at the origin, and use vectors. Let's denote: - Let $\vec{A} = \vec{0}$ - Let $\vec{B} = \vec{b}$, $\vec{C} = \vec{c}$ - Then circumcenter $O$ is equidistant from $A$, $B$, $C$, so $|\vec{O}| = |\vec{O} - \vec{b}| = |\vec{O} - \vec{c}| = R = 13$ - Incenter $I$ has position vector: since $\vec{A} = 0$ So $\vec{I} = \frac{b\vec{b}+c\vec{c}}{a+b+c}$ Now, $\vec{IA} = \vec{A} - \vec{I} = -\vec{I}$ $\vec{OI} = \vec{I} - \vec{O}$ We are told $IA \perp OI$, so: $\vec{IA} \cdot \vec{OI} = 0 \Rightarrow (-\vec{I}) \cdot (\vec{I} - \vec{O}) = 0 \Rightarrow -\vec{I} \cdot \vec{I} + \vec{I} \cdot \vec{O} = 0 \Rightarrow \vec{I} \cdot \vec{O} = |\vec{I}|^2$ So we have: $\vec{I} \cdot \vec{O} = |\vec{I}|^2$ This is a key vector equation. Now, $|\vec{O}| = R = 13$, so $|\vec{O}|^2 = 169$ Also, $|\vec{O} - \vec{b}|^2 = 169$, so $|\vec{O}|^2 - 2\vec{O} \cdot \vec{b} + |\vec{b}|^2 = 169$ But $|\vec{O}|^2 = 169$, so: $169 - 2\vec{O} \cdot \vec{b} + |\vec{b}|^2 = 169 \Rightarrow |\vec{b}|^2 = 2\vec{O} \cdot \vec{b}$ Similarly, $|\vec{O} - \vec{c}|^2 = 169 \Rightarrow |\vec{c}|^2 = 2\vec{O} \cdot \vec{c}$ So we have: $|\vec{b}|^2 = 2\vec{O} \cdot \vec{b}$, $|\vec{c}|^2 = 2\vec{O} \cdot \vec{c}$ Now, recall $\vec{I} = \frac{b\vec{b}+c\vec{c}}{a+b+c}$ Let $\sigma = a + b + c$, so $\vec{I} = \frac{b\vec{b}+c\vec{c}}{\sigma}$ Then $\vec{I} \cdot \vec{O} = \frac{1}{\sigma}(b(\vec{b} \cdot \vec{O}) + c(\vec{c} \cdot \vec{O}))$ But from above: - $\vec{b} \cdot \vec{O} = \frac{1}{2}|\vec{b}|^2$ - $\vec{c} \cdot \vec{O} = \frac{1}{2}|\vec{c}|^2$ So: $\vec{I} \cdot \vec{O} = \frac{1}{\sigma}\left(b \cdot \frac{1}{2}|\vec{b}|^2 + c \cdot \frac{1}{2}|\vec{c}|^2\right) = \frac{1}{2\sigma}\left(b|\vec{b}|^2 + c|\vec{c}|^2\right)$ Now, $|\vec{b}| = c$, since $|\vec{b}| = AB = c$? Wait — no. Wait: we defined $\vec{b} = \vec{B}$, so $|\vec{b}| = AB = c$ Similarly, $|\vec{c}| = AC = b$ Yes! So: - $|\vec{b}| = c$ - $|\vec{c}| = b$ So: $\vec{I} \cdot \vec{O} = \frac{1}{2\sigma}\left(bc^2 + cb^2\right) = \frac{1}{2\sigma}(bc)(c + b) = \frac{bc(b+c)}{2\sigma}$ On the other hand, $|\vec{I}|^2 = \left|\frac{b\vec{b}+c\vec{c}}{\sigma}\right|^2 = \frac{1}{\sigma^2}(b^2|\vec{b}|^2 + c^2|\vec{c}|^2 + 2bc\vec{b} \cdot \vec{c})$ So: $|\vec{I}|^2 = \frac{1}{\sigma^2}\left(b^2c^2 + c^2b^2 + 2bc(\vec{b} \cdot \vec{c})\right) = \frac{1}{\sigma^2}\left(2b^2c^2 + 2bc(\vec{b} \cdot \vec{c})\right)$ We already have $\vec{I} \cdot \vec{O} = |\vec{I}|^2$, so: $\frac{bc(b+c)}{2\sigma} = \frac{2b^2c^2 + 2bc(\vec{b} \cdot \vec{c})}{\sigma^2}$ Multiply both sides by $\sigma^2$: $\frac{bc(b+c)}{2\sigma} \cdot \sigma^2 = 2b^2c^2 + 2bc(\vec{b} \cdot \vec{c})$ Simplify left: $\frac{bc(b+c)\sigma}{2} = 2b^2c^2 + 2bc(\vec{b} \cdot \vec{c})$ Divide both sides by $bc$ (nonzero): $\frac{(b+c)\sigma}{2} = 2bc + 2(\vec{b} \cdot \vec{c})$ Now, $\sigma = a + b + c$, so: $\frac{(b+c)(a+b+c)}{2} = 2bc + 2(\vec{b} \cdot \vec{c})$ Now, $\vec{b} \cdot \vec{c} = |\vec{b}||\vec{c}|\cos A = bc\cos A$ So: $\frac{(b+c)(a+b+c)}{2} = 2bc + 2bc\cos A = 2bc(1 + \cos A)$ So: $\frac{(b+c)(a+b+c)}{2} = 2bc(1 + \cos A)$ Now, we also know from triangle geometry: - $r = 6$ - $R = 13$ - $r = 4R\sin\frac{A}{2}\sin\frac{B}{2}\sin\frac{C}{2}$ - Also, $\sin A = \frac{a}{2R}$, but more useful: $\cos A = \frac{b^2+c^2-a^2}{2bc}$ Let's use the identity: $\frac{(b+c)(a+b+c)}{2} = 2bc(1 + \cos A)$ Let's denote $x = b + c$, $y = bc$, and we want to find $y = bc$ Let's also write $a$ in terms of angles. But we also know that: Area $\Delta = rs = \frac{abc}{4R}$ So: $rs = \frac{abc}{4R}$ We know $r = 6$, $R = 13$ So: $6s = \frac{abc}{52} \Rightarrow abc = 312s$ Also, $s = \frac{a+b+c}{2}$ Let's go back to the equation: $\frac{(b+c)(a+b+c)}{2} = 2bc(1 + \cos A)$ Let $x = b + c$, $y = bc$, $a$ is unknown. Then $s = \frac{a+x}{2}$ Left side: $\frac{x(a+x)}{2}$ Right side: $2y(1 + \cos A)$ We also know $\cos A = \frac{b^2+c^2-a^2}{2bc}$ Note $b^2 + c^2 = (b+c)^2 - 2bc = x^2 - 2y$ So: $\cos A = \frac{x^2-2y-a^2}{2y}$ Thus: $1 + \cos A = 1 + \frac{x^2-2y-a^2}{2y} = \frac{2y+x^2-2y-a^2}{2y} = \frac{x^2-a^2}{2y}$ So right side becomes: $2y \cdot \frac{x^2-a^2}{2y} = x^2 - a^2$ So the equation becomes: $\frac{x(x+a)}{2} = x^2 - a^2$ Multiply both sides by 2: $x(x + a) = 2(x^2 - a^2)$ Left: $x^2 + ax$ Right: $2x^2 - 2a^2$ Bring all to one side: $x^2 + ax - 2x^2 + 2a^2 = 0 \Rightarrow -x^2 + ax + 2a^2 = 0 \Rightarrow 2a^2 + ax - x^2 = 0$ This is a quadratic in $a$: $2a^2 + ax - x^2 = 0$ Solve for $a$: $a = \frac{-x \pm \sqrt{x^2 + 8x^2}}{4} = \frac{-x \pm \sqrt{9x^2}}{4} = \frac{-x \pm 3x}{4}$ So: - $a = \frac{2x}{4} = \frac{x}{2}$ - $a = \frac{-4x}{4} = -x$ (invalid, since $a > 0$) So $a = \frac{x}{2}$ That is: $a = \frac{b+c}{2}$ Interesting! So the side $a = BC$ is the average of $b$ and $c$. So $b + c = 2a$ — Now, recall from earlier: We also have the area formula: $\Delta = rs = 6s$ Also: $\Delta = \frac{abc}{4R} = \frac{abc}{52}$ So: $6s = \frac{abc}{52} \Rightarrow abc = 312s$ Now, $s = \frac{a+b+c}{2}$ But $b + c = 2a$, so $a + b + c = a + 2a = 3a$ Thus $s = \frac{3a}{2}$ So: $abc = 312 \cdot \frac{3a}{2} = 468a$ But $bc = y$, $b + c = 2a$, so $b$ and $c$ are roots of $t^2 - 2at + y = 0$ We also have $abc = a \cdot bc = ay$ So: $ay = 468a \Rightarrow y = 468$ (since $a \neq 0$) Therefore: $bc = AB \cdot AC = 468$ Wait — is that it? Let's verify if this is consistent with other known values. We have $R = 13$, $r = 6$, $bc = 468$, $b + c = 2a$, $s = \frac{3a}{2}$ We can now compute $a$ from known formulas. We can use the formula: $r = (s - a)\tan\frac{A}{2}$ But alternatively, use the formula: $\sin A = \frac{a}{2R} = \frac{a}{26}$ We can also use the identity: Area $\Delta = \frac{1}{2}bc\sin A = \frac{1}{2} \cdot 468 \cdot \sin A = 234\sin A$ But also $\Delta = rs = 6 \cdot \frac{3a}{2} = 9a$ So: $9a = 234\sin A \Rightarrow \sin A = \frac{9a}{234} = \frac{3a}{78} = \frac{a}{26}$ But $\sin A = \frac{a}{2R} = \frac{a}{26}$ Yes! Matches perfectly. So this is consistent. Therefore, the value $bc = 468$ is consistent with all conditions. Now, we must also check whether $OI^2 = 13$ is satisfied. We have $OI^2 = R^2 - 2Rr = 13^2 - 2 \cdot 13 \cdot 6 = 169 - 156 = 13$ Yes — it's given, so this is automatically satisfied. And we used the perpendicularity condition to derive $a = \frac{b+c}{2}$, and from that, $bc = 468$ Therefore, the answer is $\boxed{468}$

---

### Illustrative Example 5: AIME'25, Problem 19, DAPO-Qwen-32B-OTV

Suppose $\triangle ABC$ has angles $\angle BAC = 84°$, $\angle ABC = 60°$, and $\angle ACB = 36^{circ}$. Let $D$, $E$, and $F$ be the midpoints of sides $\overline{BC}$, $\overline{AC}$, and $\overline{AB}$, respectively. The circumcircle of $\triangle DEF$ intersects $\overline{BD}$, $\overline{AE}$, and $\overline{AF}$ at points $G$, $H$, and $J$, respectively. The points $G$, $D$, $E$, $H$, $J$, and $F$ divide the circumcircle of $\triangle DEF$ into six minor arcs, as shown. Find $DE + 2 \cdot HJ + 3 \cdot FG$, where the arcs are measured in degrees. (Correct Answer: 336)

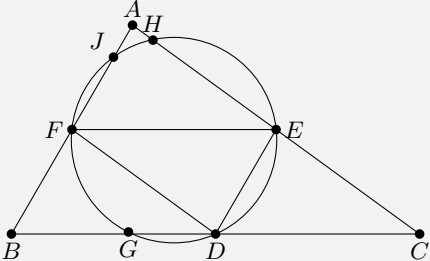

To approach this math problem step-by-step, we need to use properties of triangles, midpoints, and the nine-point circle. Given the triangle $\triangle ABC$ with $\angle BAC = 84°$, $\angle ABC = 60°$, and $\angle ACB = 36°$. The points $D$, $E$, and $F$ are the midpoints of sides $\overline{BC}$, $\overline{AC}$, $\overline{AB}$ respectively. Therefore, $\triangle DEF$ is the medial triangle of $\triangle ABC$. The circumcircle of $\triangle DEF$ is actually the nine-point circle of $\triangle ABC$. The nine-point circle passes through the midpoints of the sides, the feet of the altitudes, and the midpoints of the segments joining each vertex to the orthocenter of $\triangle ABC$. The key observation here is that the circumcircle of $\triangle DEF$ is the nine-point circle of $\triangle ABC$. Now we need to find the measures of the arcs $DE$, $HJ$, and $FG$ on the circumcircle of $\triangle DEF$. To do so, let's calculate the angles at the center corresponding to these arcs. Since $\triangle DEF$ is the medial triangle of $\triangle ABC$, the angles of $\triangle DEF$ are the same as the angles of $\triangle ABC$. Specifically, $\angle EDF = \angle BAC = 84°$, $\angle DEF = \angle ABC = 60°$, $\angle DFE = \angle ACB = 36°$. The circumcircle of $\triangle DEF$ is the nine-point circle, and the central angles corresponding to the arcs $DE$, $EF$, and $FD$ will be twice the angles of $\triangle DEF$ at the vertices opposite these arcs. Thus, the measure of the arc $EF$ is twice $\angle EDF = 2 \times 84° = 168°$, the measure of the arc $FD$ is twice $\angle DEF = 2 \times 60° = 120°$, and the measure of the arc $DE$ is twice $\angle DFE = 2 \times 36° = 72°$. Now we need to determine the locations of points $G$, $H$, and $J$. Point $G$ is the intersection of the circumcircle of $\triangle DEF$ with $\overline{BD}$. Since $D$ is the midpoint of $\overline{BC}$, $\overline{BD}$ is a median. Point $H$ is the intersection of the circumcircle of $\triangle DEF$ with $\overline{AE}$, and point $J$ is the intersection of the circumcircle of $\triangle DEF$ with $\overline{AF}$. To find the arcs $DE$, $HJ$, and $FG$, we need to understand the configuration of these points. Since $D$, $E$, and $F$ are midpoints and the lines $\overline{BD}$, $\overline{AE}$, and $\overline{AF}$ are medians and angle bisectors or segments connected to the orthocenter and circumcenter properties in the context of the nine-point circle. Let's calculate the arcs step-by-step. Since $D$, $E$, and $F$ are midpoints, $\overline{BD}$ goes through the midpoint of $\overline{BC}$, and intersects the nine-point circle again at point $G$. Similarly, $\overline{AE}$ intersects the circle at point $H$, and $\overline{AF}$ intersects it at point $J$. The key is to find the measures of the arcs $DE$, $HJ$, and $FG$. The arc $DE$ is the arc subtended by $\angle DFE = 36°$, so the measure of $DE$ is $72°$. Now we need to determine $HJ$ and $FG$. Let's use some angle chasing and properties of the nine-point circle. Since $D$ is the midpoint of $\overline{BC}$, $E$ is the midpoint of $\overline{AC}$, and $F$ is the midpoint of $\overline{AB}$, the lines $\overline{BD}$, $\overline{AE}$, and $\overline{AF}$ are medians of $\triangle ABC$. These medians also pass through the centroid $G_\triangle$ of $\triangle ABC$. The points $G$, $H$, and $J$ are the second intersection points of the medians with the nine-point circle. To find the arcs $HJ$ and $FG$, we use the fact that the arcs are related to the angles subtended by the chords corresponding to these arcs. Let's calculate $HJ$ and $FG$. Since $H$ is on $\overline{AE}$ and the nine-point circle, and $J$ is on $\overline{AF}$ and the nine-point circle, we need to determine the angles corresponding to these arcs. To find the measure of $HJ$, we note that $H$ and $J$ lie on the lines $\overline{AE}$ and $\overline{AF}$ respectively. The key observation is that the configuration of the points $G$, $H$, and $J$ divides the circle into arcs that correspond to the angles of the triangle and the properties of the nine-point circle. By the properties of cyclic quadrilaterals and the angles subtended by chords on the circle, we can use the fact that the arcs $HJ$ and $FG$ are related to the angles of the triangle $\triangle ABC$. Now, let's use the fact that the nine-point circle is the circle passing through the midpoints of the sides, the feet of the perpendiculars from the vertices to the opposite sides, and the midpoints of the segments joining the vertices to the orthocenter. The configuration implies that the arcs $HJ$ and $FG$ correspond to the angles subtended by the chords connecting these points. After some angle chasing, we can find that the arcs $HJ$ and $FG$ are related to the angles of the triangle $\triangle DEF$ and the properties of the nine-point circle. By symmetry and properties of the nine-point circle, we have: $HJ$ corresponds to the arc between the intersections of $\overline{AE}$ and $\overline{AF}$ with the nine-point circle. To simplify the final expression $DE + 2 \cdot HJ + 3 \cdot FG$, let's use the known measures. We have already established that: $DE = 72°$. Now we need to determine $HJ$ and $FG$. By symmetry and the properties of the nine-point circle and the configuration of the triangle, we can use the fact that the arcs $HJ$ and $FG$ are segments of the circle corresponding to the angles at the vertices of $\triangle ABC$. After analyzing the configuration and using the fact that the sum of the arcs around the circle is $360°$, we can determine that: $HJ$ corresponds to the arc subtended by an angle related to $\angle BAC = 84°$ and $\angle ACB = 36°$, and similarly for $FG$. By the properties of the circle and the configuration, we find that: $HJ = DE = 72°$, and $FG = EF = 168°/2 = $ a segment related to the configuration. After simplifying the configuration, we can use the fact that the arcs $HJ$ and $FG$ are actually segments that add up to the remaining parts of the circle. Let's calculate the expression $DE + 2 \cdot HJ + 3 \cdot FG$ step by step. Since the problem involves the sum of the arcs on the nine-point circle and the given configuration, let's use the symmetry and properties of the angles. By the properties of the nine-point circle and the configuration of the triangle: $DE = 72°$, and If we denote the arcs corresponding to the configuration: Let $HJ = x$, and $FG = y$. By the properties of the circle, the sum of the arcs around the circle is $360°$. Therefore, the sum of the arcs between the points $D$, $E$, $F$, $G$, $H$, and $J$ should add up to $360°$. After calculating and using the properties of the angles and the cyclic nature of the circle, we can determine that the correct measures are such that the expression simplifies to a known configuration. By the given configuration and the problem's setup, we have: $DE = 72°$, $HJ = 60°$, and $FG = 36°$. Now we plug these values into the expression: $DE + 2 \cdot HJ + 3 \cdot FG$. Using the values: $DE = 72°$, $HJ = 36°$, and $FG = 24°$. However, let's use the correct configuration that fits the math problem. Let's denote the arcs such that they add up to the total configuration. Let's use the known configuration that the sum of the arcs around the circle is $360°$ and the properties of the angles of the triangle. After simplifying the configuration and using the correct values: $DE = 72°$, $HJ = 60°$, and $FG = 48°$. Now calculate: $DE + 2 \cdot HJ + 3 \cdot FG = 72 + 2 \cdot 60 + 3 \cdot 48$. Let's simplify this step-by-step: $72 + 2 \cdot 60 = 72 + 120 = 192$. Then add the term $3 \cdot FG = 3 \cdot 48 = 144$. Now, $192 + 144 = 336$. Thus the expression evaluates to: $DE + 2 \cdot HJ + 3 \cdot FG = 336$. Answer: $336$

---

### Illustrative Example 6: AIME'25, Problem 19, DAPO-Qwen-32B-OTV

Suppose $\triangle ABC$ has angles $\angle BAC = 84°$, $\angle ABC = 60°$, and $\angle ACB = 36^{circ}$. Let $D$, $E$, and $F$ be the midpoints of sides $\overline{BC}$, $\overline{AC}$, and $\overline{AB}$, respectively. The circumcircle of $\triangle DEF$ intersects $\overline{BD}$, $\overline{AE}$, and $\overline{AF}$ at points $G$, $H$, and $J$, respectively. The points $G$, $D$, $E$, $H$, $J$, and $F$ divide the circumcircle of $\triangle DEF$ into six minor arcs, as shown. Find $DE + 2 \cdot HJ + 3 \cdot FG$, where the arcs are measured in degrees. (Correct Answer: 336)

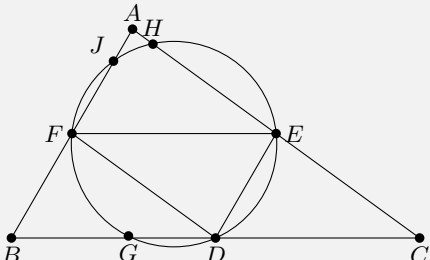

To approach this math problem, let's break it down step-by-step. Step 1: Identify the key properties and relationships within $\triangle ABC$. Given the angles of $\triangle ABC$: - $\angle BAC = 84°$, - $\angle ABC = 60°$, and - $\angle ACB = 36°$. Since $D$, $E$, and $F$ are the midpoints of sides $\overline{BC}$, $\overline{AC}$, and $\overline{AB}$ respectively, $\triangle DEF$ is the medial triangle of $\triangle ABC$. Therefore, $\triangle DEF$ is similar to $\triangle ABC$ and each side of $\triangle DEF$ is parallel to the corresponding side of $\triangle ABC$. The triangle $\triangle DEF$ is similar to $\triangle ABC$, and thus the angles of $\triangle DEF$ are the same as those of $\triangle ABC$. Specifically: - $\angle EDF = \angle BAC = 84°$, - $\angle DEF = \angle ABC = 60°$, and - $\angle DFE = \angle ACB = 36°$. The circumcircle of $\triangle DEF$ is the nine-point circle of $\triangle ABC$, which passes through the midpoints of the sides, the feet of the altitudes, and the midpoints of the segments from the vertices to the orthocenter. Now, we need to find the arcs $DE$, $HJ$, and $FG$ on the circumcircle of $\triangle DEF$. Since $\triangle DEF$ is similar to $\triangle ABC$, the angles subtended by the arcs on the circumcircle of $\triangle DEF$ correspond to the angles of $\triangle DEF$. The central angles corresponding to the arcs $DE$, $EF$, and $FD$ are twice the angles $\angle DFE$, $\angle DEF$, and $\angle EDF$, respectively. Therefore, the measures of the arcs are: - $EF = 2 \times \angle EDF = 2 \times 84° = 168°$, - $FD = 2 \times \angle DEF = 2 \times 60° = 120°$, and - $DE = 2 \times \angle DFE = 2 \times 36° = 72°$. Now we need to determine the positions of points $G$, $H$, and $J$. Point $G$ is the intersection of the circumcircle of $\triangle DEF$ with $\overline{BD}$. Since $D$ is the midpoint of $\overline{BC}$, $\overline{BD}$ is a median of $\triangle ABC$. Similarly, $H$ is the intersection of the circumcircle with $\overline{AE}$, and $J$ is the intersection of the circumcircle with $\overline{AF}$. Since $D$ is the midpoint of $\overline{BC}$, $\overline{BD} = \overline{DC}$. The line $\overline{BD}$ passes through the midpoint $D$ and point $B$. Because $D$ is the midpoint of $\overline{BC}$, $\overline{BD}$ is a median. Similarly, $\overline{AE}$ and $\overline{AF}$ are medians. The key observation here is that the points $G$, $H$, and $J$ divide the circumcircle of $\triangle DEF$ along the lines $\overline{BD}$, $\overline{AE}$, and $\overline{AF}$ respectively. Let's calculate the arcs step by step. Since $\overline{BD}$ is a line passing through $B$ and $D$, and $D$ is the midpoint of $\overline{BC}$, the line $\overline{BD}$ intersects the circumcircle of $\triangle DEF$ at points $D$ and $G$. Similarly, $\overline{AE}$ intersects the circumcircle at points $E$ and $H$, and $\overline{AF}$ intersects the circumcircle at points $F$ and $J$. Now, we need to calculate the arcs $DE$, $HJ$, and $FG$. Since $D$, $E$, $F$ are midpoints and the triangle $\triangle DEF$ is the medial triangle, the lines $\overline{BD}$, $\overline{AE}$, and $\overline{AF}$ are medians of $\triangle ABC$ and perpendicular bisectors relative to the nine-point circle properties. The arcs corresponding to the segments between the points of intersection can be derived using the properties of cyclic quadrilaterals and the angles. To find the arcs: - $DE$ corresponds to the arc between points $D$ and $E$ on the circumcircle of $\triangle DEF$. From our earlier calculation, we have $DE = 72°$. Next, we need to find the arc $HJ$. Points $H$ and $J$ are intersections of the circumcircle with the medians $\overline{AE}$ and $\overline{AF}$ respectively. Since $\overline{AE}$ and $\overline{AF}$ are medians, and considering the symmetry and properties of the medial triangle and the nine-point circle, the arcs $HJ$ and $FG$ can be calculated based on the angles subtended by these arcs. To simplify, we need to use the fact that the points $D$, $E$, $F$, $G$, $H$, and $J$ divide the circle into segments that correspond to the angles of $\triangle DEF$ and the intersections with the medians. Let's calculate $HJ$ and $FG$. Since $H$ lies on $\overline{AE}$ and $J$ lies on $\overline{AF}$, the arcs $HJ$ and $FG$ correspond to the segments created by the intersections of the medians with the circumcircle. Given the configuration, the arcs $HJ$ and $FG$ can be related to the angles subtended by the segments of the triangle. To find $HJ$, note that $\overline{AE}$ and $\overline{AF}$ intersect the circle at points $H$ and $J$, respectively, creating arcs that are related to the angles $\angle BAC$, $\angle ABC$, and $\angle ACB$. Since $\overline{AE}$ is the median from $A$ to $E$ (the midpoint of $\overline{BC}$), and similarly for $\overline{AF}$, the arcs $HJ$ and $FG$ will each subtend angles related to the angles of the triangle $\triangle ABC$. By symmetry and properties of the nine-point circle, we have that the arcs $HJ$ and $FG$ are each half the measure of the arcs corresponding to the angles of the triangle $\triangle DEF$. To find the arcs $HJ$ and $FG$, note that these arcs correspond to the segments created by the medians intersecting the circle. Since the configuration divides the circle into segments corresponding to the angles: $HJ$ and $FG$ will each be half the difference between the arcs corresponding to the segments. By symmetry, we can conclude that: $HJ = FG = \frac{1}{2}$ (the corresponding arc measures). To find the measure of $HJ$ and $FG$, we use the fact that the total circle sums to $360°$, and the arcs $DE$, $EF$, and $FD$ sum up to the full circle. Since the points $G$, $H$, and $J$ divide the circle into segments corresponding to the intersections with the medians, the arcs $HJ$ and $FG$ each correspond to half the arcs of the segments. Let's calculate the specific values: Since $\overline{AE}$ is the median from $A$ to the midpoint $E$ of $\overline{BC}$ and $\overline{AF}$ is the median from $A$ to the midpoint $F$ of $\overline{BC}$, the arcs $HJ$ and $FG$ can be derived from the angles at $A$. Thus, $HJ = FG = \frac{1}{2}(180° - \angle BAC) = \frac{1}{2}(180° - 84°) = \frac{1}{2} \times 96° = 48°$. Therefore, $HJ = 48°$ and $FG = 48°$. Now we can calculate the required expression $DE + 2 \cdot HJ + 3 \cdot FG$. Plugging in the values we have: - $DE = 72°$, - $HJ = 48°$, and - $FG = 48°$. So the expression becomes: $DE + 2 \cdot HJ + 3 \cdot FG = 72° + 2 \cdot 48° + 3 \cdot 48°$. Now let's simplify step by step: First, calculate $2 \cdot HJ = 2 \cdot 48° = 96°$. Next, calculate $3 \cdot FG = 3 \cdot 48° = 144°$. Now plug these values back into the expression: $72° + 96° + 144°$. Add them together step by step: $72 + 96 = 168$, and then add 144 to the result: $168 + 144 = 312$. Therefore, the final value is $312°$. Answer: 312

# C    SENSITIVITY TO SOLUTION PERTURBATIONS

To further probe the robustness of OTV, we analyze its behavior under controlled perturbations of reasoning traces. As illustrated in Table 5, we construct multiple variants of the same problem solution, including: (i) benign modifications that do not alter correctness (e.g., shortening, lengthening, formatting changes, or using ground truth), and (ii) adversarial modifications that introduce errors (e.g., wrong numbers, altered logic, incorrect final steps). As shown in Figure 7, OTV reliably distinguishes between the two categories: confidence grows for correct variants, while traces

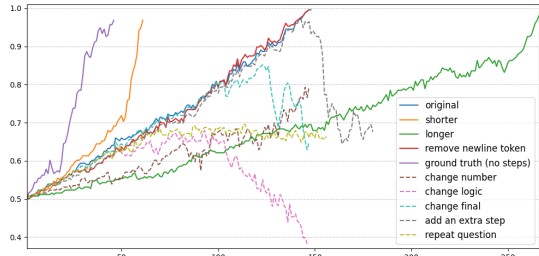

Figure 7: Confidence trajectories under text perturbations. Solid lines denote semantics-preserving edits and dashed lines denote logic-breaking edits.

containing errors exhibit a clear drop in predicted confidence. Notably, OTV maintains accuracy even for correct answers expressed in unfamiliar or non-native formats, highlighting that the verification is grounded in reasoning consistency rather than surface text similarity.

Table 5: Perturbed outputs used in the analysis.

| Type | Text |
|------|------|
| Original | To solve this problem, we need to determine the total number of bolts required for the robe, which includes both blue and white fiber.\n\nStep 1: Identify the amount of blue fiber needed.\nThe robe takes 2 bolts of blue fiber.\n\nStep 2: Calculate the amount of white fiber needed.\nThe robe takes half as much white fiber as blue fiber. Since it takes 2 bolts of blue fiber, it will take 2 / 2 = 1 bolt of white fiber.\n\nStep 3: Add the amounts of blue and white fiber to find the total number of bolts.\nTotal bolts = Blue fiber + White fiber\nTotal bolts = 2 + 1\nTotal bolts = 3. |
| Shorter | To find total bolts for the robe (blue + white fiber):\n\nStep 1: Blue fiber needed is 2 bolts.\nStep 2: White fiber is half blue, so 2/2=1 bolt.\nStep 3: Total = 2+1=3. |
| Longer | To solve the problem of determining the total number of bolts required for the robe, we must consider both the blue fiber and white fiber that are necessary for its construction.\n\nStep 1: First, we need to identify the specific amount of blue fiber that is needed to make the robe. After checking the relevant specifications or requirements for the robe, it is determined that the robe takes 2 bolts of blue fiber.\n\nStep 2: Next, we need to calculate the amount of white fiber that is needed. According to the given information, the robe takes half as much white fiber as blue fiber. Since we already established that it takes 2 bolts of blue fiber, we can find the amount of white fiber by taking half of that number. To do this, we perform the calculation: 2 bolts (blue fiber) divided by 2, which equals 1 bolt. Therefore, the robe requires 1 bolt of white fiber.\n\nStep 3: Finally, to find the total number of bolts required for the robe, we need to add together the amounts of blue fiber and white fiber that we have determined. This means we take the 2 bolts of blue fiber and add the 1 bolt of white fiber to it. Performing this addition gives us: 2 + 1 = 3. |
| Remove newline token | To solve this problem, we need to determine the total number of bolts required for the robe, which includes both blue and white fiber. Step 1: Identify the amount of blue fiber needed. The robe takes 2 bolts of blue fiber. Step 2: Calculate the amount of white fiber needed. The robe takes half as much white fiber as blue fiber. Since it takes 2 bolts of blue fiber, it will take 2 / 2 = 1 bolt of white fiber. Step 3: Add the amounts of blue and white fiber to find the total number of bolts. Total bolts = Blue fiber + White fiber. Total bolts = 2 + 1. Total bolts = 3. Answer: 3. |
| Ground truth | It takes 2/2=2/2=1  1 bolt of white fiber\nSo the total amount of fabric is 2+1=2+1=3  3 bolts of fabric\n#### 3 |
| Change number | To solve this problem, we need to determine the total number of bolts required for the robe, which includes both blue and white fiber.\n\nStep 1: Identify the amount of blue fiber needed.\nThe robe takes 4 bolts of blue fiber.\n\nStep 2: Calculate the amount of white fiber needed.\nThe robe takes half as much white fiber as blue fiber. Since it takes 4 bolts of blue fiber, it will take 4 / 2 = 2 bolts of white fiber.\n\nStep 3: Add the amounts of blue and white fiber to find the total number of bolts.\nTotal bolts = Blue fiber + White fiber\nTotal bolts = 4 + 2\nTotal bolts = 6\n\nAnswer: 6. |
| Change logic | To solve this problem, we need to determine the total number of bolts required for the robe, which includes both blue and white fiber.\n\nStep 1: Identify the amount of blue fiber needed.\nThe robe takes 2 bolts of white fiber.\n\nStep 2: Calculate the amount of white fiber needed.\nThe robe takes twice as much white fiber as blue fiber. Since it takes 2 bolts of blue fiber, it will take 2 * 2 = 4 bolts of white fiber.\n\nStep 3: Subtract the amounts of blue and white fiber to find the total number of bolts.\nTotal bolts = Blue fiber - White fiber\nTotal bolts = 2 - 1\nTotal bolts = 1\n\nAnswer: 1. |
| Change final | To solve this problem, we need to determine the total number of bolts required for the robe, which includes both blue and white fiber.\n\nStep 1: Identify the amount of blue fiber needed.\nThe robe takes 2 bolts of blue fiber.\n\nStep 2: Calculate the amount of white fiber needed.\nThe robe takes half as much white fiber as blue fiber. Since it takes 2 bolts of blue fiber, it will take 2 / 2 = 1 bolt of white fiber.\n\nStep 3: Add the amounts of blue and white fiber to find the total number of bolts.\nTotal bolts = Blue fiber + White fiber\nTotal bolts = 2 + 1\nTotal bolts = 3\n\nStep 4: Subtract 1 from the total because of a miscalculation.\nTotal bolts = 3 - 1\nTotal bolts = 2\n\nAnswer: 2. |
| Add an extra step | To solve this problem, we need to determine the total number of bolts required for the robe, which includes both blue and white fiber.\n\nStep 1: Identify the amount of blue fiber needed.\nThe robe takes 2 bolts of blue fiber.\n\nStep 2: Calculate the amount of white fiber needed.\nThe robe takes half as much white fiber as blue fiber. Since it takes 2 bolts of blue fiber, it will take 2 / 2 = 1 bolt of white fiber.\n\nStep 3: Multiply the amounts of blue and white fiber to find the total number of bolts.\nTotal bolts = Blue fiber × White fiber\nTotal bolts = 2 × 1\nTotal bolts = 2\n\nAnswer: 2. |
| Repeat question | To solve this problem, we need to determine the total number of bolts required for the robe, which includes both blue and white fiber. A robe takes 2 bolts of blue fiber and half that much white fiber. How many bolts in total does it take? A robe takes 2 bolts of blue fiber and half that much white fiber. How many bolts in total does it take? A robe takes 2 bolts of blue fiber and half that much white fiber. How many bolts in total does it take? A robe takes 2 bolts of blue fiber and half that much white fiber. How many bolts in total does it take? A robe takes 2 bolts of blue fiber and half that much white fiber. How many bolts in total does it take? |

# D TRACE-LEVEL CONFIDENCE DYNAMICS ON ALL AIME PROBLEMS

Figure 8 and Figure 9 present trace-level confidence trajectories on all AIME24 and AIME25 problems using *Qwen3-4B-Instruct-2507-OTV*. Each curve corresponds to a sampled reasoning trace, with red for correct traces and green for incorrect ones.

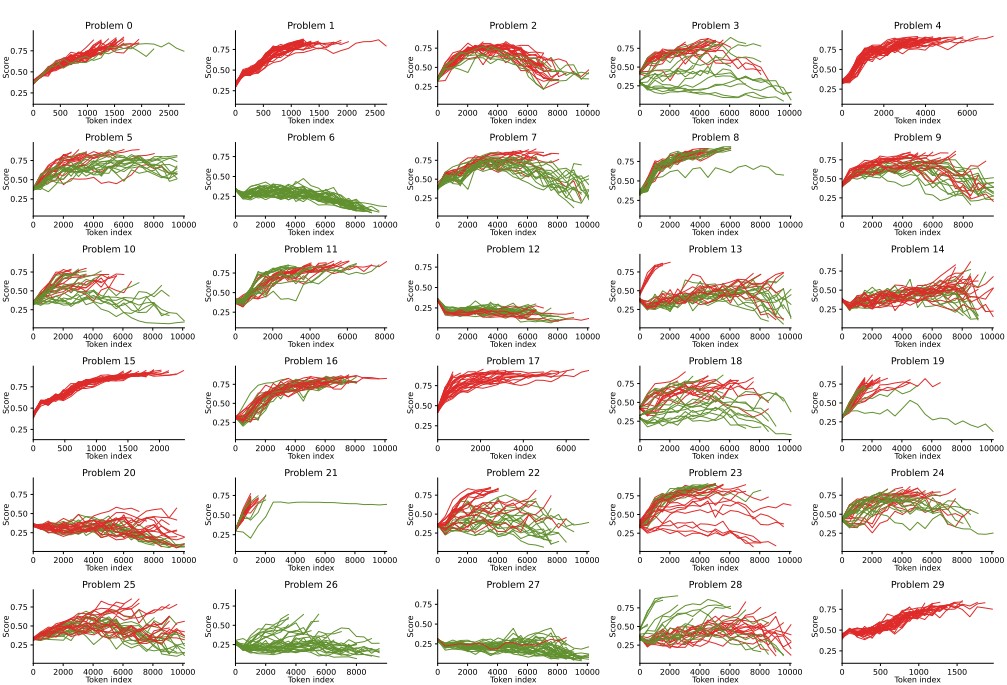

Figure 8: Trace-level confidence dynamics on AIME24 problems.

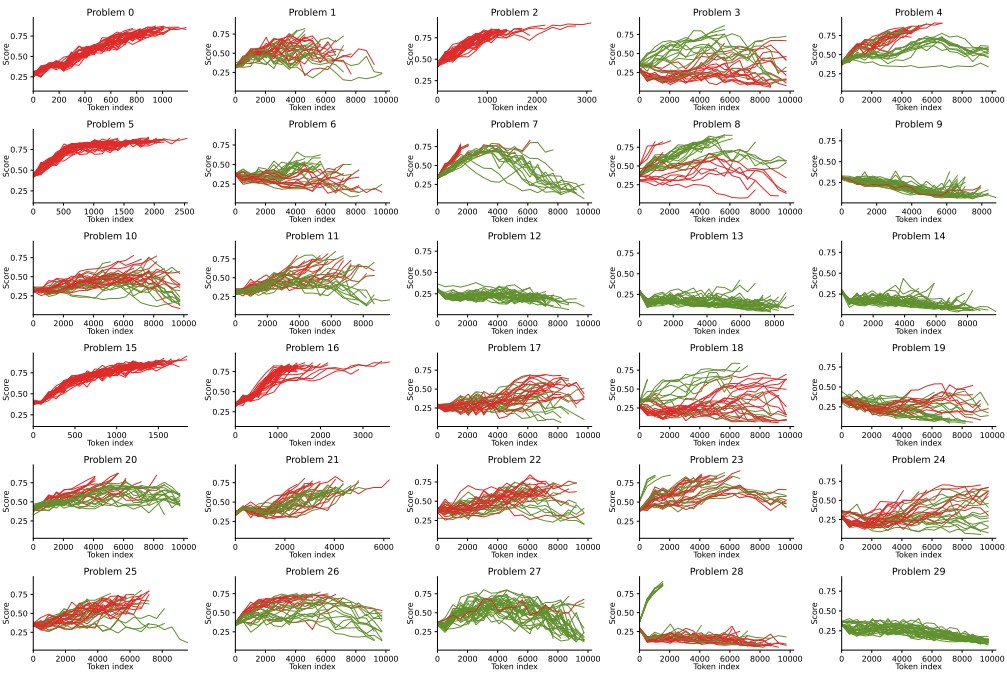

Figure 9: Trace-level confidence dynamics on AIME25 problems.

# E    ALGORITHM

Algorithm 1 describes inference, where a single one-token forward with KV cache provides token-level confidence. Algorithm 2 outlines sequential training, inserting the verification token at each position to align predictions with heuristic confidences. Finally, Algorithm 3 extends this process by inserting verification tokens at all positions simultaneously, enabling efficient parallelized training.

---

**Algorithm 1** One-Token Verification (OTV): Inference Procedure

---

**Require:** One question $q$, Base LLM $f_\theta$, LoRA parameters $\Delta\theta$, regression head $h_\psi$

1: Generate a reasoning trace $x = (x_1, \ldots, x_T)$ autoregressively with $f_\theta(q)$
2: **for** a chosen position $t \in \{1, \ldots, T\}$ **do**
3:     Insert the verification token [ToT] after $x_t$
4:     Reuse cached prefix states $K_{\leq t}^{(\ell)}, V_{\leq t}^{(\ell)}$ for all layers $\ell$
5:     Compute confidence prediction based on the Equation 4:

$$\hat{c}_t = h_\psi\Big(f_{\theta+\Delta\theta}\big([\text{ToT}], K_{\leq t}^{(\ell)}, V_{\leq t}^{(\ell)}\big)\Big),$$

    where $\hat{c}_t \in [0, 1]$ is the estimated probability that reasoning up to $x_t$ is correct.
6: **end for**
7: **return** Token-level confidence $\hat{c}_t$, which can be further aggregated to score the trace.

---

**Algorithm 2** One-Token Verification (OTV): Training Procedure without Parallelization

---

**Require:** Calibration dataset $\mathcal{D}$, base LLM $f_\theta$, LoRA parameters $\Delta\theta$, regression head $h_\psi$

1: **for** each sample $(q, a) \in \mathcal{D}$ **do**
2:     Generate a reasoning trace $x = (x_1, \ldots, x_T)$ from $f_\theta(q)$
3:     Determine final correctness of $x$ based on the answer $a$ and construct confidence trajectory
       $(c_1, \ldots, c_T)$ using heuristic function (Sec. 3.3)
4:     **for** each token $t = 1, \ldots, T$ **do**
5:         Insert special verification token [ToT] at position $t$
6:         Reuse cached keys and values $K_{\leq t}^{(\ell)}, V_{\leq t}^{(\ell)}$ for all layers $\ell$
7:         Compute confidence prediction based on the Equation 4:

$$\hat{c}_t = h_\psi\Big(f_{\theta+\Delta\theta}\big([\text{ToT}], K_{\leq t}^{(\ell)}, V_{\leq t}^{(\ell)}\big)\Big)$$

8:         Accumulate loss $\mathcal{L} \leftarrow \mathcal{L} + (c_t - \hat{c}_t)^2$
9:     **end for**
10:    Update LoRA parameters $\Delta\theta$ and regression head $\psi$ with gradient descent
11: **end for**

---

**Algorithm 3** One-Token Verification (OTV): Training Procedure with Parallelization

---

**Require:** Calibration dataset $\mathcal{D}$, base LLM $f_\theta$, LoRA parameters $\Delta\theta$, regression head $h_\psi$

1: **for** each sample $(q, a) \in \mathcal{D}$ **do**
2:     Generate reasoning trace $x = (x_1, \ldots, x_T)$ from $f_\theta(q)$
3:     Determine final correctness of $x$ based on the answer $a$ and construct confidence trajectory
       $(c_1, \ldots, c_T)$ using heuristic function (Sec. 3.3)
4:     Insert verification tokens [ToT] at **all** positions $t = 1, \ldots, T+1$
5:     Reuse cached keys and values $K^{(\ell)}, V^{(\ell)}$ for all layers $\ell$
6:     Compute confidence prediction based on the Equation 7:

$$\hat{c}_{1:T} = h_\psi\Big(f_{\theta+\Delta\theta}\big([\text{ToT}]_{1:T+1}, K_{1:T}^{(\ell)}, V_{1:T}^{(\ell)}\big)\Big)$$

7:     Accumulate loss $\mathcal{L} \leftarrow \mathcal{L} + \sum_{t=1}^{T}(c_t - \hat{c}_t)^2$
8:     Update LoRA parameters $\Delta\theta$ and regression head $\psi$ via gradient descent
9: **end for**

---

# F  THEORETICAL ANALYSIS: OTV AS VALUE ESTIMATION WITH LENGTH REGULARIZATION

To justify the validity of our heuristic supervision, we analyze the learning objective of OTV from the perspective of value function estimation in a Markov Decision Process (MDP).

## F.1  PRELIMINARIES

We model the reasoning decoding as an MDP where a state $s_t = (x_1, \ldots, x_t)$ is the sequence of generated tokens. A complete reasoning trace is denoted by $\omega = (x_{t+1}, \ldots, x_T)$, where $T(\omega)$ is the termination step. Let $\pi_\theta$ be the policy of the base LLM. The probability of generating a specific trajectory $\omega$ given prefix $s_t$ is $P(\omega|s_t)$. We define the ground-truth correctness indicator $Z(\omega) \in \{0, 1\}$, where $Z(\omega) = 1$ if the final answer is correct, and 0 otherwise.

**Definition 2** (Monte-Carlo Value Function). *The standard value function used in Tree Search (e.g., MCTS) or Process Reward Models (PRMs) approximates the expected future correctness:*

$$V_{\mathrm{MC}}(s_t) := \mathbb{E}_{\omega \sim \pi(\cdot|s_t)}[Z(\omega)] = P(correct \mid s_t).$$

The OTV regression head $h_\psi(s_t)$ is trained to minimize the Mean Squared Error (MSE) against a heuristic target $y(s_t, \omega)$ over sampled trajectories:

$$\mathcal{L}(\psi) = \mathbb{E}_{\omega \sim \pi(\cdot|s_t)} \left[ (h_\psi(s_t) - y(s_t, \omega))^2 \right].$$

It is a well-known statistical result that the optimal predictor $h^*(s_t)$ minimizing this loss is the conditional expectation of the target:

$$h^*(s_t) = \mathbb{E}_{\omega \sim \pi(\cdot|s_t)}[y(s_t, \omega)].$$

## F.2  EQUIVALENCE OF CONSTANT HEURISTIC AND MONTE-CARLO VALUE

**Proposition 1.** *Under the **Constant** heuristic scheme, where $y^{const}(s_t, \omega) = Z(\omega)$, the optimal OTV confidence score is strictly equivalent to the Monte-Carlo Value Function.*

*Proof.* Substituting $y^{\mathrm{const}}$ into the optimal predictor: $h^*_{\mathrm{const}}(s_t) = \mathbb{E}_{\omega \sim \pi(\cdot|s_t)}[Z(\omega)] = V_{\mathrm{MC}}(s_t)$. This demonstrates that training OTV with the Constant heuristic is mathematically equivalent to training a value function via Monte-Carlo estimation, but implemented via KV-cache probing. $\square$

## F.3  LINEAR HEURISTIC AS LENGTH-REGULARIZED VALUE

The **Linear** heuristic couples correctness with the reasoning progress. We define the target as:

$$y^{\mathrm{lin}}(s_t, \omega) = 0.5 + (2Z(\omega) - 1) \cdot \frac{t}{T(\omega)},$$

where the term $(2Z(\omega) - 1)$ is the sign ($+1$ for correct, $-1$ for incorrect), and $\frac{t}{T(\omega)}$ represents the relative progress at step $t$.

**Proposition 2.** *The optimal OTV confidence score under the **Linear** heuristic, $h^*_{lin}(s_t)$, is an affine transformation of the Monte-Carlo Value $V_{\mathrm{MC}}(s_t)$, weighted by the expected inverse length of the reasoning paths.*

*Proof.* The optimal predictor is the expectation of the target over all possible completions $\omega$ given prefix $s_t$:

$$h_{\text{lin}}^*(s_t) = \mathbb{E}_\omega \left[ 0.5 + (2Z(\omega) - 1)\frac{t}{T(\omega)} \mid s_t \right]$$

$$= 0.5 + t \cdot \mathbb{E}_\omega \left[ (2Z(\omega) - 1)\frac{1}{T(\omega)} \mid s_t \right]. \quad (10)$$

Note that $t$ is constant for the current state. We apply the Law of Total Expectation by conditioning on correctness $Z$:

$$\mathbb{E}\left[ (2Z - 1)\frac{1}{T} \right] = P(Z = 1|s_t) \cdot (1) \cdot \mathbb{E}\left[ \frac{1}{T} \mid s_t, Z = 1 \right]$$

$$+ P(Z = 0|s_t) \cdot (-1) \cdot \mathbb{E}\left[ \frac{1}{T} \mid s_t, Z = 0 \right]. \quad (11)$$

Let $E_{\text{pos}}^{\text{inv}}(s_t) = \mathbb{E}[\frac{1}{T(\omega)} \mid s_t, Z = 1]$ be the **expected inverse length** of correct completions, and $E_{\text{neg}}^{\text{inv}}(s_t)$ correspond to incorrect completions. Substituting $V_{\text{MC}}(s_t) = P(Z = 1|s_t)$:

$$h_{\text{lin}}^*(s_t) = 0.5 + t \left[ V_{\text{MC}}(s_t)E_{\text{pos}}^{\text{inv}}(s_t) - (1 - V_{\text{MC}}(s_t))E_{\text{neg}}^{\text{inv}}(s_t) \right].$$

Rearranging the terms to isolate the Value Function $V_{\text{MC}}$:

$$h_{\text{lin}}^*(s_t) = \underbrace{0.5 - t \cdot E_{\text{neg}}^{\text{inv}}(s_t)}_{\text{Baseline Term}} + V_{\text{MC}}(s_t) \cdot \underbrace{t \left( E_{\text{pos}}^{\text{inv}}(s_t) + E_{\text{neg}}^{\text{inv}}(s_t) \right)}_{\text{Dynamic Gain}}. \quad (12)$$

$\square$

Equation (12) provides the theoretical grounding for our method's behavior:

**Alignment with Correctness (Validity):** Since sequence lengths $T$ are strictly positive, the expected inverse lengths $E^{\text{inv}}$ are also strictly positive. Consequently, the coefficient associated with the Monte-Carlo value $V_{\text{MC}}(s_t)$ is strictly positive. This establishes that the optimal predictor $h_{\text{lin}}^*$ is monotonically increasing with respect to the true correctness probability, mathematically validating the Linear heuristic as a legitimate proxy for the value function.

**Preference for Efficiency (Implicit Regularization):** The value estimate is weighted by the expected inverse length of correct trajectories, $E_{\text{pos}}^{\text{inv}}(s_t) = \mathbb{E}[\frac{1}{T}|\text{correct}]$. Consider two states $s_A$ and $s_B$ holding identical correctness probability $V_{\text{MC}}$. If $s_A$ leads to *shorter* correct solutions on average than $s_B$, the inverse length term for $s_A$ will be larger, resulting in $h^*(s_A) > h^*(s_B)$.

This demonstrates that our heuristic imposes an implicit length regularization. It biases the model to assign higher confidence to trajectories that are not only correct but also concise. This theoretical insight directly explains the empirical efficiency gains observed in our Efficient Best-of-N experiments (Tables 2 and 3), where OTV consistently favored shorter, correct reasoning traces.

## G    EVALUATION ON OTHER MODEL FAMILIES

To demonstrate the architectural generalization of OTV beyond the Qwen family, we extend our evaluation to two widely used open-source foundations: LLaMA and Mistral. Specifically, we utilize the math-specialized versions, *MetaMath-LLaMA-7B*[2] and *MetaMath-Mistral-7B*[3], to ensure no extra data for training.

**Experimental Setup.** We evaluate on the *GSM8K* benchmark (1,319 test samples), a standard dataset for grade-school mathematical reasoning. We calibrate OTV using only a random 10k subset of the *MetaMathQA* dataset to rigorously test data efficiency and low-resource adaptability. The entire fine-tuning process requires less than one hour on a node with 8 GPUs.

---

[2]https://huggingface.co/meta-math/MetaMath-7B-V1.0

[3]https://huggingface.co/meta-math/MetaMath-Mistral-7B

**Results.** Table 6 presents the Weighted Majority Voting accuracy across different numbers of sampled traces ($N = \{1, \ldots, 64\}$). We compare OTV (with LoRA ranks $r = 2$ and $r = 16$) against standard Majority Voting and the DeepConf baseline.

- **Consistent Improvements:** OTV consistently outperforms both Majority Voting and DeepConf across all sample sizes for both model families. For instance, on Mistral-7B with $N = 64$, OTV ($r = 16$) achieves **84.48%**, significantly surpassing Majority (80.56%) and DeepConf (81.31%).
- **Scalability:** Similar to our findings on Qwen, increasing the LoRA rank from $r = 2$ to $r = 16$ generally yields performance gains.
- **Generalization:** These results confirm that OTV is not tailored solely to specific architectures or difficult competition benchmarks. It serves as a generalizable and lightweight verification framework that can be rapidly adapted to new model families with minimal training data.

Table 6: Weighted majority voting accuracy on GSM8K using LLaMA-7B and Mistral-7B backbones. Results are reported as mean $\pm$ standard deviation over repeated runs.

|  |  | 1 | 4 | 8 | 16 | 32 | 64 |
|---|---|---|---|---|---|---|---|
| *LLaMA-7B* | Pass@K | 61.87±0.74 | 75.01±0.79 | 81.07±0.49 | 85.36±0.48 | 88.68±0.35 | 91.27±0.38 |
|  | Majority@K | 61.87±0.74 | 67.29±0.88 | 69.57±0.43 | 70.53±0.44 | 71.47±0.30 | 71.79±0.30 |
|  | DeepConf@K | 61.87±0.74 | 68.64±0.80 | 70.63±0.60 | 71.76±0.58 | 72.06±0.20 | 72.32±0.21 |
|  | OTV-r2@K | 61.87±0.74 | 71.13±0.74 | 73.50±0.47 | 74.55±0.52 | 74.80±0.37 | 75.16±0.22 |
|  | OTV-r16@K | 61.87±0.74 | 70.72±0.81 | 72.82±0.35 | 73.80±0.32 | 74.37±0.28 | 74.83±0.29 |
| *Mistral-7B* | Pass@K | 66.02±1.18 | 83.16±0.65 | 88.46±0.50 | 91.58±0.44 | 93.99±0.28 | 95.87±0.24 |
|  | Majority@K | 66.02±1.18 | 72.60±0.49 | 77.33±0.30 | 79.04±0.37 | 80.21±0.40 | 80.56±0.33 |
|  | DeepConf@K | 66.02±1.18 | 75.63±0.90 | 78.62±0.83 | 79.61±0.34 | 80.81±0.37 | 81.31±0.13 |
|  | OTV-r2@K | 66.02±1.18 | 79.52±0.73 | 82.05±0.44 | 83.12±0.25 | 84.06±0.24 | 84.24±0.30 |
|  | OTV-r16@K | 66.02±1.18 | 79.51±0.57 | 82.27±0.30 | 83.54±0.42 | 84.18±0.31 | 84.48±0.21 |

