# OpenReview forum: "One-Token Verification for Reasoning LLMs, Anytime, Anywhere"
_ICLR.cc/2026/Conference — Submitted to ICLR 2026_

### Official Review · Reviewer_KSje · 2025-10-23

**Soundness:** 2
**Presentation:** 3
**Contribution:** 2
**Rating:** 4
**Confidence:** 4

**Summary:**

This paper proposes an efficient way (using an adapter, based on LoRA) to train a verifier to assess intermediate results of reasoning traces. The experimental results demonstrate the superior performance of the proposed method on math benchmarks against SOTA baselines.

**Strengths:**

1. This paper proposes a clever way to address the computational efficiency for the intermediate assessment, which could also be easily adapted to other scenarios such as actor-critic based PPO training.
2. The experimental results show a clever gain against the baselines, especially for the weighted average scenario
3. The paper is well-written and clearly presented

**Weaknesses:**

1. The second contribution in the Introduction part does not hold
> We design a heuristic supervision and parallel training scheme that provides dense step-level signals, making the proposed OTV method efficient, scalable, and adaptable to metrics beyond answer correctness

[1,2] have already pointed out the value/advantage function should be the correct training target for step-wise signals, it remains unclear to me why these heuristics are developed. There lacks the discussion and comparison to these prior works.

2. Main evaluation is only conducted on Qwen models and AIME benchmarks. Given the evaluation issues raised by recent works [3,4], I think the evaluation is not comprehensive.

3. The BoN results do not show a consistent improvement (Table 2).

[1] Feng et al. Step-by-Step Reasoning for Math Problems via Twisted Sequential Monte Carlo. ICLR 2025.

[2] Setlur et al. Rewarding Progress: Scaling Automated Process Verifiers for LLM Reasoning. ICLR 2025

[3] Shao et al. Spurious Rewards: Rethinking Training Signals in RLVR.

[4] Hochlehnert et al. A Sober Look at Progress in Language Model Reasoning: Pitfalls and Paths to Reproducibility. COLM 2025.

**Questions:**

The novelty part of this work seems not so clear. Given my list of previous works, this paper only stands out in proposing a more efficient strategy to finetune/store the verifier rather than making any algorithmic innovations. Most importantly, there lacks a clear efficiency comparison with baselines.

For example, within the same time budget, could DeepConf have more sampled trajectories?

---

> ### Author Response · Authors · 2025-11-26
> **Response to Reviewer KSje (1/2)**
>
> > [W1] The second contribution in the Introduction part does not hold. [1,2] have already pointed out the value/advantage function should be the correct training target for step-wise signals, it remains unclear to me why these heuristics are developed. There lacks the discussion and comparison to these prior works.
>
>
> **Answer:**
>
> We thank the reviewer for this critical question. While we agree with [1,2] that the value function is the ideal theoretical target, we developed heuristic supervision to address the computational bottleneck of estimating true values:
>
> 1. The "Why": Efficiency & Scalability
>     - Estimating a true step-wise value function (via Monte-Carlo) typically requires sampling $K$ rollouts for every step, resulting in prohibitive training costs ($T \times K$ complexity).
>     - We designed heuristics to enable single-pass training (1-to-1 mapping between a trace and its labels). This allows OTV to scale efficiently to large, diverse datasets (like DAPO17K) without the massive compute overhead required to construct ground-truth value trees. As noted in our manuscript, this avoids the expense of MCTS-based sampling methods.
>
> 2. The "Validity": Theoretical Equivalence
>     - Critically, these heuristics are not arbitrary. In the newly added Appendix F (summarized in the General Response), we mathematically prove that these heuristics as valid proxies for value functions.
>
> We have updated the Section 3.3 of our manuscript to explicitly discuss the MCTS-based methods [1,2], and we will highlight that OTV offers a training-efficient and theoretically-equivalent alternative to them.
>
>
> > [W2] Main evaluation is only conducted on Qwen models and AIME benchmarks. Given the evaluation issues raised by recent works [3,4], I think the evaluation is not comprehensive.
>
> **Answer:**
>
> We thank the reviewer for highlighting these recent works. We are well aware of the pitfalls in reasoning evaluation highlighted by [3] and [4], and we designed our evaluation protocol specifically to avoid them.
>
> 1. The cited works primarily critique the high variance and instability of Pass@1, where results can fluctuate wildly based on prompts or random seeds. Since our work targets parallel thinking, we naturally focused on Pass@128 and Majority@128 as the standard metrics for this setting. Crucially, to strictly address reproducibility concerns, all results are averaged over 64 different seeds and same prompts (sampling 128 traces per run). This protocol eliminates random fluctuation, directly addressing the reproducibility concerns raised in [4].
>
> 2. Regarding the "narrow coverage, we respectfully refer the reviewer to Appendix G (added in the rebuttal). We extended evaluations to LLaMA-7B and Mistral-7B families and evaluated on GSM8K (grade-school math) which has 1.3k samples for testing. Even on these different architectures and datasets, OTV consistently outperforms baselines, confirming that our method's effectiveness generalizes well beyond the Qwen family.
>
>
> > [W3] The BoN results do not show a consistent improvement (Table 2).
>
> **Answer:**
>
> We acknowledge that results vary in standard BoN. However, we emphasize two key points regarding practical utility:
>
> - Standard BoN incurs the exact same computational cost as Weighted Majority Voting. In this shared-cost setting (Table 1), OTV consistently outperforms all baselines across all models.
> - OTV demonstrates superior efficacy in Efficient BoN (e.g., HALF 300), where it achieves the best or near-best accuracy. This confirms OTV is the superior choice for both the most accurate (Weighted Majority) and most efficient (HALF 300) inference strategies.
>
>
> > [Q1] The novelty part of this work seems not so clear. Given my list of previous works, this paper only stands out in proposing a more efficient strategy to finetune/store the verifier rather than making any algorithmic innovations.
>
> **Answer:**
>
> We respectfully disagree with the assessment that OTV is merely a strategy for efficient storage. We believe **OTV introduces fundamental algorithmic innovations** in how verification is computed and parallelized within Transformer architectures:
>
> 1. OTV introduces LoRA-based Role Vectors to algorithmically switch the base model from Generator to Verifier. This allows **direct probing of the internal KV Cache** ("brain scan"), accessing reasoning dynamics that external verifiers simply cannot reach.
> 2. We designed a novel attention computation using "Lower-Left Augmentation" and custom masking (defined in Eq. 5-7). This algorithmically enables the model to verify all historical tokens simultaneously during training, solving the bottleneck of sequential verification.
> 3. Unlike external verifiers that incur $O(N^2)$ prefilling cost per step, OTV algorithmically reduces verification to $O(N)$ by reusing the pre-computed KV cache.

---

> ### Author Response · Authors · 2025-11-26
> **Response to Reviewer KSje (2/2)**
>
> > [Q2] Most importantly, there lacks a clear efficiency comparison with baselines. For example, within the same time budget, could DeepConf have more sampled trajectories?
>
> **Answer:**
>
> We appreciate this question, as it allows us to clarify the efficiency of OTV. The answer is **No**. The computational overhead of OTV is negligible compared to the cost of generating a reasoning trace, meaning DeepConf would not gain a meaningful sampling advantage within the same time budget.
>
> - In standard settings (Majority Voting and BoN), inference latency is dominated by generating reasoning traces that often span thousands of tokens. Since OTV reuses the KV cache and requires only a single-token decoding step, its overhead is practically negligible, similar to DeepConf's logit aggregation. The cost of OTV verification is too small to enable sampling even a fraction of an additional reasoning trace.
> - Even in efficient BoN variants, the cost disparity remains significant. As noted in our manuscript (Line 308), for 128 traces using the HALF 300 strategy, reasoning generation requires approximately 70,200 tokens whereas verification requires only 252 tokens. Thus, OTV does not impose a penalty that limits the sampling budget.

---

### Official Review · Reviewer_NZWw · 2025-10-29

**Soundness:** 2
**Presentation:** 3
**Contribution:** 1
**Rating:** 2
**Confidence:** 4

**Summary:**

To assess the quality of reasoning traces at any point in generation and for any token position, the authors introduce OTV, a lightweight framework for assessing reasoning quality via a single one-token forward during generation. They conduct extensive experiments across multiple reasoning LLMs and math benchmarks.

**Strengths:**

1. The paper shows that OTV can surpass recent PRMs across various LLMs and settings (e.g., self-consistency, BoN).
2. The authors visualizes the confidence trajectories of different scoring methods, highlighting that OTV more faithfully captures the
underlying reasoning quality.

**Weaknesses:**

1. The authors miss an important reference: "Token-Supervised Value Models for Enhancing Mathematical Problem-Solving Capabilities of Large Language Model, ICLR 2025". The missing reference is totally critical due to the fact that this work already offers anytime, anywhere verification even without any special token like `[ToT]`. Moreover, this work already shows its effectiveness for step-level signals, but the authors do not compare OTV with this work at all. Finally, this work theoretically shows that its method can allow tree search algorithms to be value-guided, but OTV is based on heuristic confidence. In other words, there is no theoretical analysis why OTV can work well. In this sense, the novelty of OTV seems negligible.
2. In addition to Figure 1, it would be instructive if the authors share the example of confidence decreasing for inference, whether this example really results in a wrong answer, and at which token the confidence decreases dramatically.
3. It seems necessary to show that OTV can be helpful for MCTS or Step-by-Step Beam Search to show that OTV can be also applied to advanced search strategies.

**Questions:**

Can OTV be extended to other tasks (e.g., code generation, neural theorem proving)?

---

> ### Author Response · Authors · 2025-11-26
> **Response to Reviewer NZWw (1/2)**
>
> > [W1] The authors miss an important reference: "Token-Supervised Value Models for Enhancing Mathematical Problem-Solving Capabilities of Large Language Model, ICLR 2025". The missing reference is totally critical due to the fact that this work already offers anytime, anywhere verification even without any special token like `[ToT]`. Moreover, this work already shows its effectiveness for step-level signals, but the authors do not compare OTV with this work at all. Finally, this work theoretically shows that its method can allow tree search algorithms to be value-guided, but OTV is based on heuristic confidence. In other words, there is no theoretical analysis why OTV can work well. In this sense, the novelty of OTV seems negligible.
>
> **Answer:**
> - We thank the reviewer for highlighting this important work, TVMs. We have incorporated it into the Related Work section of our revised manuscript (highlighted in blue).
> - We acknowledge distinct conceptual parallels between TVMs and OTV: both move beyond sparse outcome supervision to utilize dense, token-wise signals for "anytime" verification. However, the implementation differs fundamentally:
>     - TVMs typically apply scalar heads to the generator's local hidden states. This is structurally analogous to the 'probe' baseline in our ablation study (Figure 5). Since OTV significantly outperforms this baseline, it empirically demonstrates the advantage of OTV's global KV-probing mechanism.
>     - OTV introduces a structural innovation, which mode-switches the model to perform a "brain scan" of the pre-computed KV Cache. This allows OTV to probe global reasoning dynamics stored in memory rather than just local states.
> - Therefore, we respectfully disagree that OTV's novelty is negligible. As detailed in **Section 1 of our General Response**, OTV differs fundamentally from existing verifiers in both mechanism and efficiency.
> - Regarding theoretical grounding, we explicitly address this in the newly added Appendix F (summarized in **Section 2 of the General Response**). We provide a rigorous derivation proving that our heuristic supervision is mathematically equivalent to tree-based value estimation, effectively learning a value function with implicit length regularization, thereby theoretically justifying its effectiveness.
>
> > [W2] In addition to Figure 1, it would be instructive if the authors share the example of confidence decreasing for inference, whether this example really results in a wrong answer, and at which token the confidence decreases dramatically.
>
> **Answer:**
> We appreciate this instructive suggestion. While Figure 1 is intended as a simplified schematic of the mechanism, we fully agree that visualizing confidence drops is essential for understanding OTV's behavior. To this end, we have provided detailed analyses in Figure 4 and Appendices B, C, and D. Notably, Appendix C (Figure 7) explicitly illustrates the dramatic confidence drops at tokens where logical errors occur, offering the concrete visualization recommended.
>
> > [W3] It seems necessary to show that OTV can be helpful for MCTS or Step-by-Step Beam Search to show that OTV can be also applied to advanced search strategies.
>
> **Answer:**
> We appreciate the suggestion to demonstrate OTV in advanced search settings. We clarify that our Efficient Best-of-N variants (e.g., HALF 300) effectively serve as a specialized Step-by-Step Search. Furthermore, we are actively exploring Tree Search integration following the reviewer's recommended literature.
>
> - Mechanistically, our "Efficient Variants" (Table 2) perform step-wise verification and pruning, dynamically allocating compute to high-scoring paths while discarding unpromising ones. This is functionally equivalent to a Beam Search under a strict token budget.
> - In our preliminary explorations, standard step-by-step Beam Search underperformed compared to our Efficient BoN strategy under identical compute budgets. We attribute this to diversity issues: beam search often suffers from mode collapse (generating highly similar reasoning chains). In contrast, Efficient BoN (sampling then pruning) maintains the high diversity crucial for complex reasoning, utilizing OTV's "anytime" signals to ensure quality without sacrificing coverage.
> - Ongoing Work on Tree Search: We acknowledge the valuable insight from the recommended work Token-Supervised Value Models, which notes that verifiers originally designed for BoN may require specific adaptations for tree search. Inspired by this work, we are currently reproducing the proposed tree search algorithms to better adapt OTV for such strategies. While our primary contribution focuses on the efficiency of KV-cache probing in BoN settings, we are actively conducting these additional experiments and will update the manuscript with any new findings during the discussion period. For the current scope, we advocate for Efficient BoN as the optimal strategy that balances OTV's speed with the need for diverse reasoning paths.

---

> ### Author Response · Authors · 2025-11-26
> **Response to Reviewer NZWw (2/2)**
>
> > [Q1] Can OTV be extended to other tasks (e.g., code generation, neural theorem proving)?
>
> **Answer:**
>
> Yes, absolutely. We appreciate the suggestion. Theoretically, OTV is applicable to any domain with verifiable outcomes.
>
> In this work, our primary focus is on Reasoning LLMs, where mathematical problem-solving capabilities serves as the primary proxy for evaluation. Consequently, our current evaluation covers the full spectrum of reasoning depth, ranging from fundamental grade-school math (GSM8K, evaluated on Qwen-3 in Section 5.2 and LLaMA/Mistral-7B in the newly added Appendix G) to complex competition-level problems (AIME, evaluated on Qwen-3 and DAPO-Qwen). We also plan to expand OTV to broader verifiable domains in future work.

---

### Official Review · Reviewer_PtHd · 2025-10-31

**Soundness:** 3
**Presentation:** 3
**Contribution:** 2
**Rating:** 4
**Confidence:** 3

**Summary:**

The paper propose One-Token Verification (OTV), a very lightweight method to assess reasoning quality during the model reasoning process. OTV trains a LoRA for self-verification as a role vector and uses special token [ToT] to measure the verification confidence. The experiment shows promising result of OTV in parallel thinking.

**Strengths:**

1. Paper is relatively well written and easy to follow.
2. OTV is sound and well constructed in the methodology.

**Weaknesses:**

1. The concept of OTV is not super new in that it uses (an adapted) model internal confidence to measure the certainty of a verifiable result.
2. Evaluation is not comprehensive enough. The evaluation only shows result against math reasoning, and have not shown OTV can generalize beyond the standard math benchmarks.
3. Related works lack the various works related to model confidence measure and early-stopping in LLM reasoning.

**Questions:**

1. Why choosing the DAPO17K dataset as the dataset to train against?
2. How will this method perform against non-mathematical datasets?
3. The experiment & Appendix D shows the various traces of problem (with correct vs incorrect) and the change of the score. Statistically, do you observe the answer flipping phenomenon to happen, meaning the retained cases actually contains the right answer (in the trajectory) but then flipped to the wrong one in later stage?

---

> ### Author Response · Authors · 2025-11-26
> **Response to Reviewer PtHd (1/2)**
>
> We sincerely thank the reviewer for the thoughtful feedback. We've addressed your concerns below.
>
> > [W1] The concept of OTV is not super new in that it uses (an adapted) model internal confidence to measure the certainty of a verifiable result.
>
> **Answer:**
>
> We respectfully clarify that OTV is technically distinct from prior internal methods in both mechanism and depth. Unlike approaches relying on shallow output probabilities (e.g., DeepConf) or self-prompting (GenRM), OTV is the first to employ a learnable LoRA Role Vector to directly probe the full reasoning dynamics stored in the KV Cache via a single token. This design is non-trivial, requiring a dedicated parallelization scheme (Eq. 7) to ensure efficiency. Empirically, OTV not only significantly outperforms these internal baselines but also demonstrates strong scalability in our ablation studies (Figure 5), where increasing the LoRA rank yields consistent performance gains.
>
>
>
> > [W2] Evaluation is not comprehensive enough. The evaluation only shows result against math reasoning, and have not shown OTV can generalize beyond the standard math benchmarks.
> > [Q1] Why choosing the DAPO17K dataset as the dataset to train against?
> > [Q2] How will this method perform against non-mathematical datasets?
>
> **Answers to [W2], [Q1] & [Q2]:**
>
> Our primary focus is on Reasoning LLMs, where mathematics serves as the "gold standard" for evaluating rigorous multi-step logic due to its objective correctness.
>
> - **Breadth of Difficulty**: We cover the full spectrum of reasoning depth, ranging from fundamental grade-school math (GSM8K, evaluated on Qwen-3 in Section 5.2 and LLaMA/Mistral in new added Appendix G) to complex competition-level problems (AIME, evaluated on Qwen-3 and DAPO-Qwen).
> - **Rationale for DAPO17K (Addressing Q1)**: We selected datasets based on standard community practices and difficulty alignment. MetaMathQA (primarily for SFT) and DAPO17K (primarily for RL) are the two dominant open-source datasets in this field, both exceeding 10k monthly downloads on HuggingFace. While we utilized both in our experiments, DAPO17K was chosen for the primary Qwen/AIME evaluations because its higher difficulty aligns better with the advanced capabilities of the Qwen family and the complexity of the AIME benchmark, thereby more effectively demonstrating the value of our method. Specifically, DAPO17K features significantly longer reasoning chains (average >5,000 tokens) compared to MetaMathQA (average ~200 tokens); this long-context density allows the verifier to learn robust, step-by-step logic discrimination rather than simple answer memorization, and our results confirm that OTV achieves significant gains on these challenging tasks.
> - **Future Scope (Addressing Q2)**: While extending OTV to domains such as code generation and output safety is a priority for future work, establishing efficacy on math is the critical first step.
>
>
> > [W3] Related works lack the various works related to model confidence measure and early-stopping in LLM reasoning.
>
> **Answer:**
>
> We thank the reviewer for the suggestion. In our original manuscript (Section 2), we have discussed key internal confidence methods (e.g., DeepConf, self-calibration, probing hidden states, GenRM) and early-stopping strategies (e.g., trajectory pruning in Line 100). We will further expand this section in the final version to include a broader range of recent studies to ensure a comprehensive overview.

---

> ### Author Response · Authors · 2025-11-26
> **Response to Reviewer PtHd (2/2)**
>
> > [Q3] The experiment & Appendix D shows the various traces of problem (with correct vs incorrect) and the change of the score. Statistically, do you observe the answer flipping phenomenon to happen, meaning the retained cases actually contains the right answer (in the trajectory) but then flipped to the wrong one in later stage?
>
> **Answer:**
>
> We appreciate this insightful question regarding the interpretability of confidence trajectories.
> - **Absence of "Answer Flipping"**: Statistically, while we frequently observe "rise-then-fall" confidence patterns, our inspection in AIME and GSM8K experiments suggests that these traces do not typically represent an answer flipping phenomenon, where retained cases containing the correct reasoning are discarded for a wrong one. In instances where the confidence was high before dropping, we found that while the trajectory might produce intermediate numbers resembling the right answer, the underlying reasoning logic had often already diverged or become incoherent. The subsequent drop in confidence at a later stage likely reflects OTV detecting this logical inconsistency, rather than rejecting a correct path.
> - **Analysis of Confidence Patterns**: Through further analysis of "rise-then-fall" versus "immediate drop" patterns, we identified several characteristic behaviors linked to reasoning quality, though these represent just a few among many observed patterns:
>     - We observed that confidence tends to drop when the model skips necessary reasoning steps or outputs intermediate results without explicit formula derivation. Conversely, confidence often recovers when the model resumes detailed, coherent derivation.
>     - For hard problems that are essentially solvable (Pass@K > 0), incorrect traces often exhibit the "rise-then-fall" pattern. The model may start with a plausible approach (high score) but stumble as complexity increases, leading to a penalty in the later stage.
>     - For simpler problems in GSM8K or clearly unsolvable problems, OTV acts decisively. We observed that confidence typically drops immediately within the first few tokens and remains suppressed, indicating early error detection rather than a late flip.
> - **Corroboration via Perturbation**: This dynamic is also supported by our  experiments in Appendix C. When we manually inject logic errors into a valid trace (e.g., the "Change logic" or "Change final" perturbations), the confidence trajectory shifts from a monotonic rise to a "rise-then-fall" shape. This demonstrates that the score drop is a specific reaction to the introduction of logical flaws.
>
> In summary, OTV functions primarily as a coherence detector. A drop in score typically signifies the detection of a logical break or step-skipping, ensuring that high confidence is generally reserved for traces that maintain consistency from start to finish.

---

### Official Review · Reviewer_DkwK · 2025-11-01

**Soundness:** 3
**Presentation:** 3
**Contribution:** 3
**Rating:** 4
**Confidence:** 4

**Summary:**

The manuscript proposes One-Token Verification (OTV), which uses a LoRA to mode-switch the LLM to switch to the “role” of a verifier. By feedforwarding a special verification token, the model can probe the base model’s KV cache and output a token level confidence score. For training signals, heuristic pseudo-labels are used for confidence trajectories. The authors also propose a method to perform parallel verification, making it possible to score all positions in one pass. On AIME datasets, OTV improves majority-voting/Best-of-N and outperforms Reward Models, and shows best or near-best performance for efficient variants of BoN.

**Strengths:**

- Does not require a fully separate verifier model, saving memory and compute.
- LoRA instead of full finetuning for the “verifier role” can mitigate catastrophic forgetting on the verifier’s side.
- Efficient parallel verification is possible.

**Weaknesses:**

- Confidence labels are synthetic and heuristic, which can drift significantly from actual process correctness. Also, as far as I know, the heuristic label scheme cannot model situations where the model succeeds in a major self-correction during its trajectory.
- Although the manuscript briefly claims to justify the fairness of its evaluation protocol, I believe it is unfair. OTV is tuned with DAPO17K for each base backbone model, but other baseline Reward Models are off-the-shelf. Thus, OTV is tailored for each idiosyncratic backbone, while other baselines are fixed to a single backbone. Also, the baseline RMs are built on backbones that are older than Qwen 3, which is the backbone for OTV.
- Experiments are conducted on only a single model family(Qwen 3) and single benchmark family (AIME 24, 25), which is arguably a narrow coverage.

**Questions:**

- How does OTV fare for the setting where the generator and the verifier are heterogeneous (e.g. OTV is built on Qwen 3 and the generator is built on Gemma 3)?

---

> ### Author Response · Authors · 2025-11-26
> **Response to Reviewer DkwK (1/2)**
>
> We sincerely thank the reviewer for the careful reading and constructive feedback. Below, we address the raised concerns, supported by our newly added theoretical analysis and extensive new experiments.
>
> > [W1] Confidence labels are synthetic and heuristic, which can drift significantly from actual process correctness. Also, as far as I know, the heuristic label scheme cannot model situations where the model succeeds in a major self-correction during its trajectory.
>
> **Answers:**
>
> We thank the reviewer for this insightful question. While heuristic labels are synthetic, we justify them as a necessary efficiency trade-off supported by theoretical grounding.
> - **Efficiency Trade-off**: Obtaining ground-truth process values via Monte-Carlo estimation is prohibitively expensive ($T \times K$ rollouts). OTV prioritizes breadth (learning from diverse problems with single rollouts) over depth (perfectly scoring a single trace).
> - **Theoretical Validity**: In Appendix F, we have added a theoretical proof demonstrating that these heuristics are valid proxies. Specifically, we show that the Constant heuristic is mathematically equivalent to the Monte-Carlo value function $\mathbb{E}[Z|s]$, while the Linear heuristic acts as a value function with implicit length regularization.
> - **Modeling Self-Correction**: It is crucial to distinguish between the target label and the learned behavior. While the target trajectory is monotonic, OTV learns to detect internal state shifts:
>     - As shown in Appendix B Example 4, when the model exhibits uncertainty (e.g., "Not sure" or "But..." ), OTV’s predicted confidence captures these internal state shifts by fluctuating, rather than blindly increasing.
>     - Furthermore, in Appendix C, when we adversarially perturb a correct solution (e.g., "Change logic" or "Change number"), OTV's confidence drops significantly despite similar text lengths. Conversely, it remains robust under benign paraphrasing ("Shorter/Longer" variants).
>
> We believe these points clarify that our heuristics are a theoretically grounded design choice to maximize training efficiency.
>
>
> > [W2] Although the manuscript briefly claims to justify the fairness of its evaluation protocol, I believe it is unfair. OTV is tuned with DAPO17K for each base backbone model, but other baseline Reward Models are off-the-shelf. Thus, OTV is tailored for each idiosyncratic backbone, while other baselines are fixed to a single backbone. Also, the baseline RMs are built on backbones that are older than Qwen 3, which is the backbone for OTV.
>
> **Answer:**
>
> We acknowledge the reviewer's accurate assessment: OTV is indeed adapted to specific backbones via LoRA, whereas the baselines are typically fully trained, off-the-shelf models. However, we believe this distinction highlights OTV's efficiency and structural superiority rather than constituting an unfair comparison.
>
> 1. Standard PRMs (e.g., Qwen2.5-PRM) typically necessitate massive datasets (e.g., ~800K samples) and full fine-tuning. In contrast, OTV achieves superior results using only DAPO17K for calibration and LoRA. This proves that "tailoring" is a low-cost and practical feature.
> 2. To address concerns regarding reliance on newer backbones (like Qwen 3), we tested Qwen-DAPO (based on the older Qwen-32B architecture). Using only its original training data (no new external data), OTV still improved performance by ~4% (AIME24) and ~8% (AIME25) over Maj@128. This confirms OTV provides structural improvements by unlocking internal verification capabilities, regardless of backbone generation or extra data.
> 3. As further detailed in our response to [W3], where we demonstrate similar gains across other models without adding new data, OTV provides consistent structural improvement with minimal fine-tuning costs.
>
> Therefore, we believe this represents a "fair" and robust validation of the method's effectiveness. Furthermore, our goal is to provide OTV for mainstream open-source models, effectively establishing them as readily available, "off-the-shelf" resources for the community.

---

> ### Author Response · Authors · 2025-11-26
> **Response to Reviewer DkwK (2/2)**
>
> > [W3]  Experiments are conducted on only a single model family (Qwen 3) and a single benchmark family (AIME 24, 25), which is arguably a narrow coverage.
>
> **Answer:**
>
> We thank the reviewer for this suggestion. While Section 5.2 of the original manuscript included evaluations on GSM8K, we agree that demonstrating broader architectural generalization is essential.
>
> In response, we have added Appendix G with new experiments on two distinct model families: **LLaMA-7B** and **Mistral-7B**. To maintain strict data controls without introducing new external sources, we applied OTV to MetaMath-LLaMA-7B and MetaMath-Mistral-7B. Crucially, to demonstrate extreme data efficiency, we did not use the full training set; instead, we calibrated OTV using only a random 10k subset of the MetaMathQA dataset.
>
>
> As shown in the Table 6, OTV consistently outperforms baselines. For instance, on Mistral-7B ($N=64$), OTV ($r=16$) achieves 84.48%, significantly surpassing Majority Voting (80.56%) and DeepConf (81.31%).
>
>
> > [Q1] How does OTV fare for the setting where the generator and the verifier are heterogeneous (e.g. OTV is built on Qwen 3 and the generator is built on Gemma 3)?
>
> **Answer:**
>
> We clarify that this setting is technically inapplicable by design. As detailed in our General Response, OTV is not an external judge processing text output; it is a LoRA-based adapter that mode-switches the generator to **probe its own internal KV cache** (analogous to a "brain scan"). Therefore, the verifier must be structurally identical to the generator to access these internal states. This tight coupling is precisely what enables our verification efficiency by eliminating the prefilling overhead required by external methods.

---

### Author Response · Authors · 2025-11-26
**General Response to All Reviewers**

Dear Reviewers and ACs:

We sincerely appreciate the diligent efforts and insightful comments from all reviewers and Area Chairs. We have observed a shared concern regarding the novelty and theoretical grounding of OTV. We believe there is a fundamental misunderstanding regarding the technical depth and distinct mechanism of our contribution, which we aim to clarify below.

1. Core Novelty: Internal "Brain Scans" vs. External "Reading"
    - **Mechanism Distinction (Internal vs. External)**: Existing methods, including PRMs, ORMs, and recent token-wise value models, treat LLM generation primarily as text input to be processed—a category we define as "external" approaches in our manuscript. In contrast, OTV is the first framework to employ a learnable LoRA-based Role Vector that mode-switches the base model, enabling it to **directly probe its own KV Cache**.
    - **The "Brain Scan" Advantage**: Analogously, if external verifiers "read" a student's written answer to grade it, OTV performs a "brain scan" of the student’s active memory (KV Cache) during the thought process. While prior internal methods exist (e.g., DeepConf, hidden states probing, generative verifiers), they are typically training-free and limited to shallow signals. OTV distinguishes itself by being explicitly trainable and leveraging the complete reasoning dynamics stored in the cache. Our experiments demonstrate that this deeper integration yields consistent performance gains over these internal baselines.
    - **Computational Efficiency ($O(N)$ vs. $O(N^2)$)**: Another benefit of reusing the pre-computed KV cache is the elimination of prefilling overhead. Verifying a step with OTV requires only a single decoding step (a forward pass of the [ToT] token). Conversely, external verifiers must re-process the sequence history, incurring $O(N^2)$ complexity relative to the context length.
    - **Training Efficiency**: OTV requires only lightweight, SFT-level optimization via LoRA. Notably, even compared to some RLVR methods that similarly rely solely on verified outcome rewards, OTV demonstrates superior performance gains over baselines while avoiding the computational complexity.

2. For theoretical grounding, we address the concerns by demonstrating in the newly added Appendix F that our heuristics function as valid and computationally efficient proxies for value functions:
    - **Constant Heuristic $\approx$ Monte-Carlo Value**: We prove that minimizing MSE against a constant outcome label $Z$ is mathematically equivalent to learning the Monte-Carlo Value $\mathbb{E}[Z|s]$.
    - **Linear Heuristic is Length-Regularized Value**: We prove that the Linear heuristic acts as a value function weighted by the inverse reasoning length. This introduces an implicit bias, encouraging the model to favor trajectories that are not only correct but also concise.
    - **The Efficiency Trade-off**: Calculating ground-truth values via Monte-Carlo requires $K$ rollouts per step. OTV prioritizes breadth (learning from thousands of diverse problems in DAPO17K with single traces) over depth (perfectly scoring one tree), offering a trade-off for verification training.

While we recognize the initial ratings were low, we earnestly hope that the clarifications regarding our core novelty along with the new extensive experiments, allow the reviewers to recognize the contribution and practical value of our work. **We remain fully available to address any further questions.** If our response has resolved the concerns, we respectfully ask that reviewers reconsider the score.

---

### Comment · Area_Chair_x1Eh · 2025-11-28

Dear Reviewers,

Thank you for your time and efforts in serving as a reviewer. The authors have submitted their rebuttal, and this AC kindly asks you to review their response and assess whether your comments have been adequately addressed.

If you have not yet done so, please raise any remaining questions by adding comments and initiating discussion as needed for points that require further clarification.

ICLR encourages reviewers to actively engage in the discussion phase, so your prompt actions are especially valuable. Thank you very much for your continued efforts and valuable contributions.

Best regards,
Your AC

---

### Meta-Review · Area_Chair_APMG · 2026-01-06

**Summary:**

The paper proposes One-Token Verification (OTV), which verifies reasoning quality via a one-token forward process during generation. The paper shows that OTV can effectively enhance reasoning performance at a relatively cheap cost. The reviewers acknowledged that the paper is well-written and can be easily extended to various scenarios. However, the reviewers raised concerns about novelty and limited evaluation.

While the rebuttal provides new empirical results and clarifications, the concerns related to novelty and limited evaluation are difficult to fully address through short comments and revisions. The authors are encouraged to revise the paper with a more extensive evaluation beyond math reasoning benchmark and to clearly demonstrate the novelty of OTV compared to closely related works. Overall, the reviewers' rating leans toward rejection. Therefore, I can't recommend the paper for acceptance at this stage.

**Reviewer Concerns:**

Resolved concerns
- Reviewer NZWw
  - No example for confidence decrease

Remaining concerns
- Reviewer DkwK
  - Confidence labels are synthetic and heuristic
  - Fair comparison issue
  - Limited evaluation setting
- Reviewer PtHd
  - Limited novelty
  - Limited evaluation
- Reviewer NZWw
  - Contribution overlaps with prior work `Token-Supervised Value Models for Enhancing...`
- Reviewer KSje
  - Limited evaluation
  - Improvement is not consistent in Table 2

**Reviewer Scores:**

- Reviewer DkwK: Would maintain the current score.
- Reviewer PtHd: Would maintain the current score.
- Reviewer NZWw: Would maintain the current score.
- Reviewer KSje: Would maintain the current score.

---

### Decision · Program_Chairs · 2026-01-26

Reject